# Pareto Set Learning for
# Expensive Multi-Objective Optimization

**Xi Lin, Zhiyuan Yang, Xiaoyuan Zhang, Qingfu Zhang**
Department of Computer Science, City University of Hong Kong
{xi.lin, zhiyuyang4-c, xzhang2523-c}@my.cityu.edu.hk, qingfu.zhang@cityu.edu.hk

## Abstract

Expensive multi-objective optimization problems can be found in many real-world applications, where their objective function evaluations involve expensive computations or physical experiments. It is desirable to obtain an approximate Pareto front with a limited evaluation budget. Multi-objective Bayesian optimization (MOBO) has been widely used for finding a finite set of Pareto optimal solutions. However, it is well-known that the whole Pareto set is on a continuous manifold and can contain infinite solutions. The structural properties of the Pareto set are not well exploited in existing MOBO methods, and the finite-set approximation may not contain the most preferred solution(s) for decision-makers. This paper develops a novel learning-based method to approximate the whole Pareto set for MOBO, which generalizes the decomposition-based multi-objective optimization algorithm (MOEA/D) from finite populations to models. We design a simple and powerful acquisition search method based on the learned Pareto set, which naturally supports batch evaluation. In addition, with our proposed model, decision-makers can readily explore any trade-off area in the approximate Pareto set for flexible decision-making. This work represents the first attempt to model the Pareto set for expensive multi-objective optimization. Experimental results on different synthetic and real-world problems demonstrate the effectiveness of our proposed method.

## 1 Introduction

Many real-world applications involve optimizing multiple costly-to-evaluate and potentially competing objectives, such as finding strong yet ductile material [38], building a neural network with high accuracy and low latency [28], and improving the quality while minimizing total charge in particle accelerator tuning [73]. Very often, these objectives conflict each other and cannot be optimized simultaneously by a single solution. Instead, there is a set of solutions with different optimal trade-offs among the objectives, called the Pareto set. For a Pareto optimal solution, none of its objective values can be further improved without deteriorating others. In addition, the evaluation of each solution could require time-consuming computation or costly physical ex-

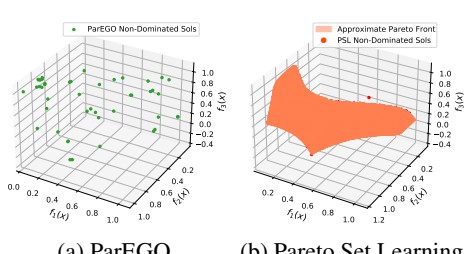

(a) ParEGO     (b) Pareto Set Learning

Figure 1: **Pareto Set Learning** can approximate the whole Pareto set, and let decision-makers easily explore any trade-off among objectives to choose their preferred solutions.

periments, and thus a large number of evaluations are unbearable. Different multi-objective Bayesian optimization (MOBO) algorithms [44, 48, 43], typically directly generalized from the single-objective Bayesian optimization (BO) [60, 39, 11, 77, 29], have been proposed to find a small set of approximate Pareto solutions with a small amount of objective function evaluation budget.

36th Conference on Neural Information Processing Systems (NeurIPS 2022).

The finite set approximation has some undesirable drawbacks. For a nontrivial multi-objective optimization problem, the Pareto set is on a continuous manifold and has infinite solutions with different optimal trade-offs among the objectives [58]. This Pareto set structure could be helpful to better select candidate solutions for expensive evaluation, and hence accelerate the optimization process of MOBO. In addition, a small set of solutions may not contain the one(s) that exactly satisfy the decision-maker's preferences. Finding the most preferred trade-off solution(s) could require several rounds of interaction with the decision-makers. These approaches would be extremely time-consuming, especially when the optimization modeler and the decision-maker are not in the same team, which is common in real-world MOBO applications [56].

This paper proposes a novel Pareto set learning (PSL) method to approximate the whole Pareto set for expensive multi-objective optimization problems with a limited evaluation budget. Our proposed method can accelerate the multi-objective Bayesian optimization process, and also provide decision-makers with more useful information to support flexible decision-making. To the best of our knowledge, this is the first attempt to learn the whole Pareto set for expensive multi-objective optimization. Our main contributions include:

- We propose a novel set model to map any trade-off preference to its corresponding Pareto solution, along with a surrogate model-based method to approximate the whole Pareto set with a limited evaluation budget.

- We develop a lightweight yet powerful batch acquisition search method for efficient MOBO, which can outperform other MOBO approaches in terms of both performance and computational cost. We demonstrate that the learned Pareto set can support flexible user-involved decision-making.

- We test our proposed method on both synthetic benchmarks and real-world application problems. The results validate the efficiency and usefulness of PSL.

## 2 Related work

**Bayesian Optimization.** Surrogate model-based methods have been widely used and studied for expensive optimization [47, 40, 65, 79]. These methods iteratively build a surrogate model to approximate the black-box objective function, and uses an acquisition function to search for the optimal solution. Much effort has been made on various design issues in Bayesian optimization, such as acquisition functions [81], high-dimensional optimization [91, 92], batch evaluation [22], scalable optimization [80, 27], and theoretical analysis [42]. Most work for BO are on single-objective optimization. We refer readers to Garnett [30] for a comprehensive introduction.

**Multi-Objective Bayesian Optimization.** MOBO extends single-objective Bayesian optimization for solving expensive multi-objective optimization problems. Although the Pareto set could contain infinite solutions, the MOBO methods typically focus on finding a single or a finite set of solutions. The scalarization-based algorithms, such as ParEGO [45] and TS-TCH [62] iteratively scalarize the multi-objective problem into single-objective ones with random preferences, and then apply single-objective BO to solve them. MOEA/D-EGO [99] adopts the MOEA/D framework [97] to solve a set of surrogate scalarized subproblems simultaneously. SMS-EGO [67] and PAL [104, 105] generalize the upper confidence bound (UCB) to multi-objective optimization. Emmerich et al. [26] and Emmerich and Klinkenberg [25] propose the probability of improvement (PI) and expected improvement (EI) for multi-objective hypervolume. Entropy search methods [33, 35, 37, 90] have also been studied in multi-objective optimization [34, 6, 84]. Bradford et al. [10] and Belakaria et al. [7] consider Thompson sampling and uncertainty maximization for multi-objective optimization, respectively. Different algorithms can be hybridized to achieve better performances [82].

Different new developments have been recently proposed to handle the issues of diverse batch evaluation [53], efficient hypervolume improvement calculation [14], noisy optimization [15, 16], high-dimensional optimization [17], and decision criteria beyond Pareto optimality [56]. Some attempts have been made to incorporate the decision-maker's preference into MOBO [3, 62, 4]. They typically need the decision-maker's preference before or during optimization, which may not always be available in real-world applications. All these MOBO methods aim to provide a finite set of approximate Pareto optimal solutions to decision-makers.

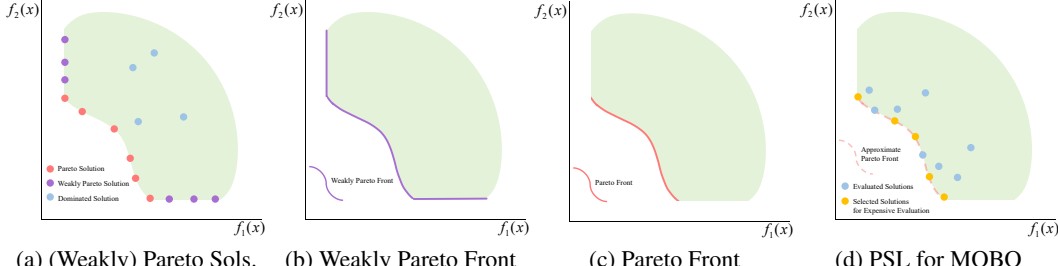

| (a) (Weakly) Pareto Sols. | (b) Weakly Pareto Front | (c) Pareto Front | (d) PSL for MOBO |

Figure 2: **(Weakly) Pareto solutions and (Weakly) Pareto front. (a)** Examples of Pareto solutions, weakly Pareto solutions, and dominated solutions. The Pareto solutions are also weakly Pareto optimal. The weakly Pareto optimal but not Pareto optimal solutions (e.g., purple points) are dominated but not strictly dominated by at least one Pareto solution. **(b)** The weakly Pareto front $\boldsymbol{f}(\mathcal{M}_{\text{weak}})$ is the image of all weakly Pareto optimal solutions $\mathcal{M}_{\text{weak}}$ in the objective space. **(c)** The Pareto front $\boldsymbol{f}(\mathcal{M}_{\text{ps}})$ is the image of all Pareto optimal solution $\mathcal{M}_{\text{ps}}$ (Pareto set) in the objective space. **(d)** Our proposed method approximates the whole Pareto set and uses it to select solutions for expensive evaluation, which can improve the search efficiency of multi-objective Bayesian optimization.

**Structure Learning.** In addition to the surrogate objective model, some methods have been proposed to explore the problem structure for Bayesian optimization. Sener and Koltun [76] learn the geometric structure of the problem in an online manner to accelerate optimization. Wang et al. [89] and Zhao et al. [101] apply Monte Carlo tree search (MCTS) to divide the search space for efficient modeling and searching. Novel latent space modelings [32, 86] have been proposed for optimization problems with complex solution representations.

**From Population to Pareto Set Learning.** For the last several decades, most multi-objective optimization methods have focused on finding a single or a finite set of Pareto optimal solutions (e.g., a population) to approximate the Pareto set [58, 24]. A few attempts have been made to approximate the whole Pareto set with simple mathematical models [68, 36, 98, 31]. Pirotta et al. [66] and Parisi et al. [63] have proposed to conduct Pareto manifold approximation for multi-objective reinforcement learning. Recently, different approaches have also been investigated to incorporate the preference information into deep neural networks for image style transfer [78, 23], multi-task learning with finite solutions [75, 49, 55, 54] or approximate Pareto front [50, 61, 74], reinforcement learning [96, 1, 2], and neural combinatorial optimization [51]. In this work, we generalize the decomposition-based multi-objective optimization algorithm (MOEA/D) [97], and propose to learn a set model which maps all valid trade-off preferences to the Pareto set for efficient MOBO.

## 3 Expensive multi-objective optimization

We consider the following expensive continuous multi-objective optimization problem:

$$\min_{\boldsymbol{x} \in \mathcal{X}} \boldsymbol{f}(\boldsymbol{x}) = (f_1(\boldsymbol{x}), f_2(\boldsymbol{x}), \cdots, f_m(\boldsymbol{x})), \tag{1}$$

where $\boldsymbol{x}$ is a solution in the decision space $\mathcal{X} \subset \mathbb{R}^n$, $\boldsymbol{f} : \mathcal{X} \to \mathbb{R}^m$ is an $m$-dimensional vector-valued objective function, and the evaluation is expensive for all individual objectives $f_i(\boldsymbol{x})$, $i = 1, \ldots, m$. For a non-trivial problem, no single solution can optimize all objectives at the same time, and we have to make a trade-off among them. We have the following definitions for multi-objective optimization:

**Definition 1 (Pareto Dominance and Strict Dominance)** *Let $\boldsymbol{x}^a, \boldsymbol{x}^b \in \mathcal{X}$, $\boldsymbol{x}^a$ is said to dominate $\boldsymbol{x}^b$, denoted as $\boldsymbol{x}^a \prec \boldsymbol{x}^b$, if and only if $f_i(\boldsymbol{x}^a) \leq f_i(\boldsymbol{x}^b), \forall i \in \{1, ..., m\}$ and $\exists j \in \{1, ..., m\}$ such that $f_j(\boldsymbol{x}^a) < f_j(\boldsymbol{x}^{(b)})$. In addition, $\boldsymbol{x}^a$ is said to strictly dominate $\boldsymbol{x}^b$ ($\boldsymbol{x}^a \prec_{strict} \boldsymbol{x}^b$), if and only if $f_i(\boldsymbol{x}^a) < f_i(\boldsymbol{x}^b), \forall i \in \{1, ..., m\}$.*

**Definition 2 (Pareto Optimality)** *A solution $\boldsymbol{x}^* \in \mathcal{X}$ is Pareto optimal if there is no $\hat{\boldsymbol{x}} \in \mathcal{X}$ such that $\hat{\boldsymbol{x}} \prec \boldsymbol{x}^*$. A solution $\boldsymbol{x}' \in \mathcal{X}$ is weakly Pareto optimal if there is no $\hat{\boldsymbol{x}} \in \mathcal{X}$ such that $\hat{\boldsymbol{x}} \prec_{strict} \boldsymbol{x}'$.*

**Definition 3 (Pareto Set/Front)** *The set of all Pareto optimal solutions $\mathcal{M}_{ps} \subseteq \mathcal{X}$ is called the Pareto set, and its image in the objective space $\boldsymbol{f}(\mathcal{M}_{ps}) = \{\boldsymbol{f}(\boldsymbol{x}) | \boldsymbol{x} \in \mathcal{M}_{ps}\}$ is called the Pareto front. Similarly, we can define the weakly Pareto set $\mathcal{M}_{weak}$ and weakly Pareto front $\boldsymbol{f}(\mathcal{M}_{weak})$.*

The strict dominance relation is stronger than the Pareto dominance since it requires strictly better values for all objectives. Therefore, the set of weakly Pareto optimal solutions $\mathcal{M}_{\text{weak}}$ (e.g., the solutions that are *not* strictly dominated) would be larger than $\mathcal{M}_{\text{ps}}$, and it is straightforward to see $\mathcal{M}_{\text{ps}} \subseteq \mathcal{M}_{\text{weak}}$. The illustration of (weakly) Pareto solution and Pareto front is shown in Figure 2.

Each Pareto solution $\boldsymbol{x} \in \mathcal{M}_{\text{ps}}$ represents a different optimal trade-off among the objectives for problem (1). Under mild conditions, the Pareto set $\mathcal{M}_{\text{ps}}$ and Pareto front $\boldsymbol{f}(\mathcal{M}_{\text{ps}})$ are both on $(m-1)$-dimensional manifold in the decision space $\mathcal{X} \in \mathbb{R}^n$ and objective space $\mathbb{R}^m$, respectively [36, 98]. The number of Pareto solutions could be infinite (i.e. $|\mathcal{M}_{\text{ps}}| = \infty$).

**Bayesian Optimization (BO)** is a model-based method for solving expensive black-box optimization problems. Given a set of already-evaluated solutions $\{\boldsymbol{X}, \boldsymbol{y}\}$, BO builds surrogate models (e.g., Gaussian process) for each objective, and defines acquisition function(s) to leverage the surrogate objective values for navigating the search space. Only promising solutions will be selected for expensive evaluation. We refer interesting reader to [30] for a detailed introduction.

**Pareto Set Learning.** The current MOBO methods aim to find a small set of finite solutions $\mathcal{S} = \{\bar{\boldsymbol{x}}^{(1)}, \bar{\boldsymbol{x}}^{(2)}, \cdots, \bar{\boldsymbol{x}}^{(|\mathcal{S}|)}\}$ to approximate the Pareto set $\mathcal{M}_{\text{ps}}$. In addition to the evaluated solutions $\mathcal{S}$, our proposed Pareto set learning (PSL) method also learns an estimated Pareto set $\mathcal{M}_{\text{psl}}$ with the predicted Pareto front $\hat{\boldsymbol{f}}(\mathcal{M}_{\text{psl}})$ to approximate the Pareto set $\mathcal{M}_{\text{ps}}$ and Pareto front $\boldsymbol{f}(\mathcal{M}_{\text{ps}})$. The whole approximate Pareto set can be easily explored by adjusting the trade-off preference as illustrated in Figure 3. With the learned Pareto set, we also develop an efficient batched solution selection approach for efficient MOBO, which will be introduced in the next section.

## 4 Pareto Set Learning for MOBO

### 4.1 Pareto set model

As pointed out in Section 3, the Pareto set can contain infinite solutions with different trade-offs. In addition, there is no complete order among the Pareto solutions. A Pareto set model for MOBO should be powerful enough to approximate the whole Pareto set, and convenient enough to easily explore any trade-off solutions. In this work, we propose to build a set model that maps any trade-off preferences to their corresponding Pareto solutions with scalarization.

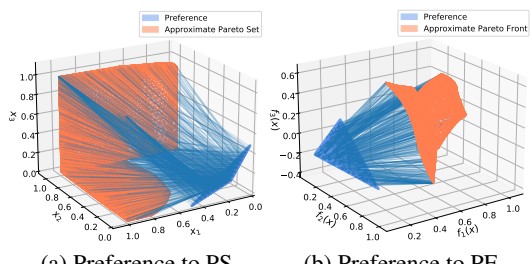

|  |  |
|---|---|
| (a) Preference to PS | (b) Preference to PF |

Figure 3: **Mapping from Preferences to Approximate Pareto Set/Front:** Our proposed PSL method learns the connection from a set of valid (infinite) trade-off preference $\Lambda = \{\boldsymbol{\lambda} \in \mathbb{R}_+^m | \sum \lambda_i = 1\}$ to **(a)** the approximate Pareto set $\mathcal{M}_{\text{psl}}$ and hence **(b)** the corresponding predicted Pareto front $\hat{\boldsymbol{f}}(\mathcal{M}_{\text{psl}})$. The whole Pareto set (front) can be easily explored by adjusting the preference. The preference simplex have been resized and rotated for better visualization.

**Scalarization.** The scalarization method provides a natural connection from a set of preferences $\Lambda = \{\boldsymbol{\lambda} \in \mathbb{R}_+^m | \sum \lambda_i = 1\}$ among the $m$ objectives to the Pareto set $\mathcal{M}_{\text{ps}}$. The most simple and straightforward approach is the weight-sum scalarization:

$$\min_{\boldsymbol{x} \in \mathcal{X}} g_{\text{ws}}(\boldsymbol{x}|\boldsymbol{\lambda}) = \min_{\boldsymbol{x} \in \mathcal{X}} \sum_{i=1}^{m} \lambda_i f_i(\boldsymbol{x}). \quad (2)$$

However, this method can only find the convex hull of Pareto front $\boldsymbol{f}(\mathcal{M}_{\text{ps}})$ [9, 24]. In this work, we use the following weighted Tchebycheff approach:

$$\min_{\boldsymbol{x} \in \mathcal{X}} g_{\text{tch}}(\boldsymbol{x}|\boldsymbol{\lambda}) = \min_{\boldsymbol{x} \in \mathcal{X}} \max_{1 \leq i \leq m} \{\lambda_i(f_i(\boldsymbol{x}) - (z_i^* - \varepsilon))\}, \quad (3)$$

where $\boldsymbol{z}^* = (z_1^*, \cdots, z_m^*)$ is the ideal vector for the objective vector $\boldsymbol{f}(\boldsymbol{x})$ (i.e. lower-bound for minimization problem), $\varepsilon > 0$ is a small positive scalar, and $u_i = (z_i^* - \varepsilon)$ is called an (unachievable) utopia value for the $i$-th objective $f_i(\boldsymbol{x})$. This scalarization method has a promising property:

**Theorem 1 (Choo and Atkins [13]).** *A feasible solution $\boldsymbol{x} \in \mathcal{X}$ is weakly Pareto optimal if and only if there is a weight vector $\boldsymbol{\lambda} > 0$ such that $\boldsymbol{x}$ is an optimal solution of the problem (3).*

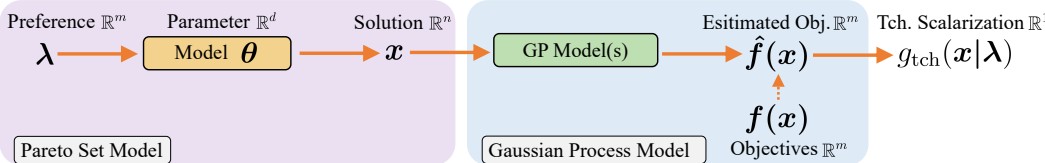

Figure 4: **Pareto Set Learning for Multi-Objective Bayesian Optimization: (a)** The Pareto set model learns a parameterized mapping from any valid preference $\boldsymbol{\lambda} \in \Lambda = \{\boldsymbol{\lambda} \in \mathbb{R}_+^m | \sum \lambda_i = 1\}$ to its corresponding solution $\boldsymbol{x}(\boldsymbol{\lambda}) \in \mathbb{R}^n$. **(b)** We build independent Gaussian process models for each objective function. With these surrogate models, the set model can be efficiently trained to approximate the Pareto set. **(c)** In this work, we use the augmented Tchebycheff scalarization to connect each preference to its corresponding Pareto solution.

In other words, all Pareto solutions $\boldsymbol{x} \in \mathcal{M}_{ps}$ can be found by solving the Tchebycheff scalarized subproblem (3) with a specific (but unknown) trade-off preference $\boldsymbol{\lambda}$. We let $\mathcal{M}_{tch}$ be the solution set for problem (3) with all valid preferences $\Lambda$ and have $\mathcal{M}_{ps} \subseteq \mathcal{M}_{weak} = \mathcal{M}_{tch}$. The weakly Pareto optimal but not Pareto optimal solutions ($\mathcal{M}_{weak} \setminus \mathcal{M}_{ps}$) are dominated (but not strictly dominated) by some Pareto solutions, and are usually not desirable for decision-making. They can be further avoided by the augmented Tchebycheff approach [83, 41]. In this work, we use the following scalarization:

$$g_{\text{tch\_aug}}(\boldsymbol{x}|\boldsymbol{\lambda}) = \max_{1 \leq i \leq m} \{\lambda_i(f_i(\boldsymbol{x}) - (z_i^* - \varepsilon))\} + \rho \sum_{i=1}^m \lambda_i f_i(\boldsymbol{x}), \quad \forall \boldsymbol{\lambda} \in \Lambda, \tag{4}$$

where $\rho$ is a sufficiently small positive scalar depends on the problem and current solution location. This form of augmentation has also been used in ParEGO [45]. With the augmentation term, the weakly dominated solutions will have larger scalarized values than the corresponding Pareto solutions in (4), and will ultimately be eliminated with the optimization process (e.g., $\mathcal{M}_{\text{tch\_aug}} = \mathcal{M}_{ps}$). In this work, we simply set $\rho = 0.001$, dynamically update $z_i^*$ as the current best value for each objective and let $\boldsymbol{\varepsilon} = 0.1|\boldsymbol{z}^*|$. This setting is robust for all problems we considered. The traditional methods focus on solving the scalarization problem (4) with a finite set of different preferences $\boldsymbol{\lambda}$ in a sequential [45] or collaborative manner [97, 99].

**Set Model.** With augmented Tchebycheff scalarization, we propose to build a set model for mapping preferences to their solutions:

$$\boldsymbol{x}(\boldsymbol{\lambda}) = h_{\boldsymbol{\theta}}(\boldsymbol{\lambda}), \tag{5}$$

where $\boldsymbol{\lambda}$ is any valid preference in $\Lambda = \{\boldsymbol{\lambda} \in \mathbb{R}_+^m | \sum \lambda_i = 1\}$, $\boldsymbol{x}(\boldsymbol{\lambda}) \in \mathcal{X}$ is its corresponding Pareto solution, and $h_{\boldsymbol{\theta}}(\boldsymbol{\lambda})$ is the Pareto set model with parameter $\boldsymbol{\theta}$. The input preference $\boldsymbol{\lambda}$ has $(m-1)$ degree of freedom, and the output solution set $\mathcal{M}_{psl} = \{\boldsymbol{x} = h_{\boldsymbol{\theta}}(\boldsymbol{\lambda}) | \boldsymbol{\lambda} \in \Lambda\}$ is on an $(m-1)$-dimensional manifold in $\mathcal{X} \in \mathbb{R}^n$. In other words, the set model maps the $(m-1)$-dimensional regular preference simplex $\Lambda$ to the $(m-1)$-dimensional solution set $\mathcal{M}_{psl}$ with complicated structure.

We want to find the optimal parameters $\boldsymbol{\theta}^*$ such that the generated set $\mathcal{M}_{psl}$ matches the solution set for augmented Tchebycheff scalarization $\mathcal{M}_{\text{tch\_aug}} = \{\boldsymbol{x}^*(\boldsymbol{\lambda}) | \boldsymbol{\lambda} \in \Lambda\}$, where

$$\boldsymbol{x}^*(\boldsymbol{\lambda}) = h_{\boldsymbol{\theta}^*}(\boldsymbol{\lambda}) = \arg\min_{\boldsymbol{x} \in \mathcal{X}} g_{\text{tch\_aug}}(\boldsymbol{x}|\boldsymbol{\lambda}), \forall \boldsymbol{\lambda} \in \Lambda. \tag{6}$$

The learned mapping is illustrated in Figure 3. Once the connection is learned, we can explore the whole approximate Pareto set/front by simply adjusting the preferences among objectives. We use an MLP neural network as the set model for all MOBO problems, which is good at capturing complicated problem structures [76]. The model details can be found in Appendix D.

### 4.2 Pareto Set Learning with Gaussian Process

Since the evaluation of $\boldsymbol{f}(\boldsymbol{x}(\boldsymbol{\lambda})) = \boldsymbol{f}(h_{\boldsymbol{\theta}}(\boldsymbol{\lambda}))$ is expensive, we use the surrogate model-based approach to learn the Pareto set model $h_{\boldsymbol{\theta}}(\boldsymbol{\lambda})$ as shown in Figure 4. Our method is orthogonal to the choice of surrogate models, and we build independent Gaussian process models for each objective as in other MOBO methods [14, 53].

**Gaussian Process Model.** A single-objective Gaussian process [69] has a prior distribution defined on the function space:

$$f(\boldsymbol{x}) \sim GP(\mu(\boldsymbol{x}), k(\boldsymbol{x}, \boldsymbol{x})), \tag{7}$$

where $\mu : \mathcal{X} \to \mathbb{R}$ is the mean function and $k : \mathcal{X} \times \mathcal{X} \to \mathbb{R}$ is the covariance kernel function. With $n$ evaluated solutions $\{\boldsymbol{X}, \boldsymbol{y}\} = \{[\boldsymbol{x}^{(i)}], [f(\boldsymbol{x}^{(i)})|i = 1, \ldots, n)\}$, the posterior distribution can be updated by maximizing the marginal likelihood based on the data. For a new solution $\boldsymbol{x}^{n+1}$, the posterior mean and variance are:

$$\hat{\mu}(\boldsymbol{x}^{(n+1)}) = \mu(\boldsymbol{x}^{(n+1)}) + \boldsymbol{k}^T \boldsymbol{K}^{-1} \boldsymbol{y}, \quad \hat{\sigma}^2(\boldsymbol{x}^{(n+1)}) = k(\boldsymbol{x}^{(n+1)}, \boldsymbol{x}^{(n+1)}) - \boldsymbol{k}^T \boldsymbol{K}^{-1} \boldsymbol{k}, \tag{8}$$

where $\boldsymbol{k} = k(\boldsymbol{x}, \boldsymbol{X})$ is the kernel vector, $\boldsymbol{K} = k(\boldsymbol{X}, \boldsymbol{X})$ is the kernel matrix, Matérn 5/2 kernel are used for all models in this work. For $m$ independent GP models, we let $\hat{\boldsymbol{\mu}}(\boldsymbol{x}) = [\hat{\mu}_1(\boldsymbol{x}), \cdots, \hat{\mu}_m(\boldsymbol{x})]$ and $\hat{\boldsymbol{\sigma}}^2(\boldsymbol{x}) = [\hat{\sigma}_1^2(\boldsymbol{x}), \cdots, \hat{\sigma}_m^2(\boldsymbol{x})]$ be the predicted mean and variance for the objective vector. Suppose we have a learned Pareto set $\mathcal{M}_{\text{psl}}$, the GP models give us both predicted value $\hat{\boldsymbol{\mu}}(\mathcal{M}_{\text{psl}}) = \{\hat{\boldsymbol{\mu}}(\boldsymbol{x}) | \boldsymbol{x} \in \mathcal{M}_{\text{psl}}\}$ and uncertainty $\hat{\boldsymbol{\sigma}}^2(\mathcal{M}_{\text{psl}}) = \{\hat{\boldsymbol{\sigma}}^2(\boldsymbol{x}) | \boldsymbol{x} \in \mathcal{M}_{\text{psl}}\}$ for the whole approximate Pareto set.

**Pareto Set Learning.** Now we propose an efficient algorithm to find the optimal parameter $\boldsymbol{\theta}^*$ for the Pareto set model $h_{\boldsymbol{\theta}}(\boldsymbol{\lambda})$. The optimal solution set $\mathcal{M}_{\text{tch\_aug}}$ for augmented Tchebycheff scalarization (4) is unknown, hence we need to optimize all solutions generated by our model $\boldsymbol{x}(\boldsymbol{\lambda}) = h_{\boldsymbol{\theta}}(\boldsymbol{\lambda})$ with respect to their corresponding augmented Tchebycheff scalarization subproblems for all valid preferences:

| **Algorithm 1** PSL with GP Models |
| :--- |
| 1: **Input:** Model $\boldsymbol{x}(\boldsymbol{\lambda}) = h_{\boldsymbol{\theta}}(\boldsymbol{\lambda})$ |
| 2: Initialize the parameters $\boldsymbol{\theta}_0$ |
| 3: **for** $t = 1$ to $T$ **do** |
| 4:     Sample preferences $\{\boldsymbol{\lambda}_k\}_{k=1}^K \sim \Lambda$ |
| 5:     Update $\boldsymbol{\theta}_t$ with gradient descent in (10) |
| 6: **end for** |
| 7: **Output:** Model $\boldsymbol{x}(\boldsymbol{\lambda}) = h_{\boldsymbol{\theta}_T}(\boldsymbol{\lambda})$ |

$$\boldsymbol{\theta}^* = \arg\min_{\boldsymbol{\theta}} \mathbb{E}_{\boldsymbol{\lambda} \sim \Lambda} g_{\text{tch\_aug}}(\boldsymbol{x} = h_{\boldsymbol{\theta}}(\boldsymbol{\lambda}) | \boldsymbol{\lambda}). \tag{9}$$

If the model is perfectly learned, the obtained approximate Pareto set $\mathcal{M}_{\text{psl}} = \{\boldsymbol{x} = h_{\boldsymbol{\theta}}(\boldsymbol{\lambda}) | \boldsymbol{\lambda} \in \Lambda\}$ should be the same as $\mathcal{M}_{\text{tch\_aug}}$. However, it is difficult to directly optimize (9) due to the expectation over infinite preferences ($|\boldsymbol{\Lambda}| = \infty$). We use Monte Carlo sampling and gradient descent to iteratively learn the model with the surrogate model:

$$\boldsymbol{\theta}_{t+1} = \boldsymbol{\theta}_t - \eta \sum_{k=1}^K \nabla_{\boldsymbol{\theta}} \hat{g}_{\text{tch\_aug}}(\boldsymbol{x} = h_{\boldsymbol{\theta}}(\boldsymbol{\lambda}_k) | \boldsymbol{\lambda}_k), \tag{10}$$

where we randomly sample $K = 10$ different valid preferences $\{\boldsymbol{\lambda}_1, \cdots, \boldsymbol{\lambda}_K\} \sim \Lambda$ at each iteration in this work. Here $\hat{g}_{\text{tch\_aug}}(\cdot)$ is the augmented Tchebycheff scalarization with predicted objective values:

$$\hat{g}_{\text{tch\_aug}}(\boldsymbol{x} | \boldsymbol{\lambda}) = \max_{1 \leq i \leq m} \{\lambda_i (\hat{f}_i(\boldsymbol{x}) - (z_i^* - \varepsilon))\} + \rho \sum_{i=1}^m \lambda_i \hat{f}_i(\boldsymbol{x}). \tag{11}$$

One design issue left is how to set the surrogate objective $\hat{\boldsymbol{f}}(\boldsymbol{x})$. If we only want to obtain the current predictive Pareto front, it is straightforward to use the posterior mean as the surrogate value. The approximate Pareto front under the posterior mean could provide valuable information to decision-makers. However, for Bayesian optimization, we have to take the uncertainty into account to balance exploitation and exploration. Many widely-used criteria, such as expected improvement (EI) [59] and upper confidence bound (UCB) [81], could be a more reasonable choice. In this work, we use the lower confidence bound (LCB) for minimization problems.

$$\hat{\boldsymbol{f}}(\boldsymbol{x}) = \hat{\boldsymbol{\mu}}(\boldsymbol{x}) - \beta \hat{\boldsymbol{\sigma}}(\boldsymbol{x}). \tag{12}$$

We simply set $\beta = \frac{1}{2}$ and discuss the performance with other surrogate values in Appendix F.9.

The expensive objective function $\boldsymbol{f}(\boldsymbol{x})$ is usually black-box and non-differentiable, but we can easily obtain the gradients for the Gaussian process and the set model. Indeed, gradient-based methods have been widely used for optimizing the acquisition function in both BO [95, 93] and MOBO [14, 53]. The max operator in Tchebycheff scalarization is technically only subdifferentiable, but it is known to have good subgradients [94] for surrogate optimization and can preserve convexity if the objectives $\{\hat{f}_i(\boldsymbol{x})\}_{i=1}^m$ are all convex [9].

The Pareto set learning algorithm with Gaussian process models is summarized in **Algorithm 1**. We find that the simple random initialization and gradient descent are enough to learn a good Pareto set approximation. The overparameterized neural network could be beneficial to overcome potential non-convexity [52].

### 4.3 Batched selection on approximate Pareto set

---

**Algorithm 2** MOBO with PSL

1: **Input:** Black-box vector-valued function $\boldsymbol{f}(\boldsymbol{x})$
2: Initial Samples $\{\boldsymbol{X}_0, \boldsymbol{y}_0\}$
3: **for** $t = 1$ to $T$ **do**
4:  Train GPs based on $\{\boldsymbol{X}_{t-1}, \boldsymbol{y}_{t-1}\}$
5:  Learn set model $h_{\boldsymbol{\theta}_t}(\boldsymbol{\lambda})$ with GPs (**Alg. 1**)
6:  Select $\{\boldsymbol{x}^{(b)}\}_{b=1}^B$ with the set model (**Alg. 3**)
7:  $\boldsymbol{X}_t \leftarrow \boldsymbol{X}_{t-1} \cup \{\boldsymbol{x}^{(b)}\}_{b=1}^B$,
   $\boldsymbol{y}_t \leftarrow \boldsymbol{y}_{t-1} \cup \boldsymbol{f}(\{\boldsymbol{x}^{(b)}\}_{b=1}^B)$
8: **end for**
9: **Output:** $\{\boldsymbol{X}_t, \boldsymbol{y}_t\}$ and final set model $h_{\boldsymbol{\theta}_T}(\boldsymbol{\lambda})$

---

**Algorithm 3** Batch Selection with PSL

1: **Input:** Model $\boldsymbol{x}(\boldsymbol{\lambda}) = \boldsymbol{h}_\theta(\boldsymbol{\lambda})$, Batch Size $B$
2: Sample $P$ preferences $\{\boldsymbol{\lambda}^{(p)}\}_{p=1}^P \sim \Lambda$
3: Generate solutions $\boldsymbol{X} = \{\boldsymbol{x}(\boldsymbol{\lambda}^{(p)})\}_{p=1}^P$ on $\mathcal{M}_{\mathrm{psl}}$
4: Find subset $\{\boldsymbol{x}^{(b)}\}_{b=1}^B \subset \boldsymbol{X}$ that has the highest HVI($\hat{\boldsymbol{f}}(\{\boldsymbol{x}^{(b)}\}_{b=1}^B)$)
5: **Output:** Batch solutions $\{\boldsymbol{x}^{(b)}\}_{b=1}^B$

---

In this subsection, we propose a lightweight yet efficient batched acquisition search for MOBO with the learned Pareto set model. The algorithm framework is shown in **Algorithm 2**. The crucial difference with other MOBO approaches is that we build a set model at each iteration for batched solution selection as shown in **Algorithm 1** and **Algorithm 3**. The batched selection procedure contains two closely related steps:

**Batch Sampling on Approximate Pareto Set.** Our model naturally supports generating an arbitrary number of solutions in batch. If the decision-maker's preferences are available, we can use preference-based sampling in this step. In this work, without any prior knowledge, we uniformly sample $P$ valid preferences $\{\boldsymbol{\lambda}^{(p)}\}_{p=1}^P$, and generate the corresponding solutions $\boldsymbol{X} = \{\boldsymbol{x}(\boldsymbol{\lambda}^{(p)})\}_{p=1}^P$ on the approximate Pareto set $\mathcal{M}_{\mathrm{psl}}$.

**Batch Selection.** At each iteration of MOBO, we typically select a small number $B$ (e.g., 5) of solutions $\boldsymbol{X}_B = \{\boldsymbol{x}^{(b)}\}_{b=1}^B$ from the sampled solutions $\boldsymbol{X}$ for expensive evaluations. To take all already evaluated solutions $\{\boldsymbol{X}_{t-1}, \boldsymbol{y}_{t-1}\}$ into consideration, we use the hypervolume [103] as the selection criteria. The hypervolume $\mathrm{HV}(\boldsymbol{y}) = \mathbf{Vol}(\boldsymbol{S})$ measures the volume of $\boldsymbol{S}$ dominated by a set $\boldsymbol{y}$ in the objective space:

$$\boldsymbol{S} = \{r \in \mathbb{R}^m \mid \exists y \in \boldsymbol{y} \text{ such that } y \prec r \prec r^*\}, \tag{13}$$

where $r^*$ is a reference point that dominated by all $y \in \boldsymbol{y}$. The hypervolume improvement (HVI) of a set $\boldsymbol{X}_B$ with respect to the already evaluated solutions $\{\boldsymbol{X}_{t-1}, \boldsymbol{y}_{t-1}\}$ can be defined as:

$$\mathrm{HVI}(\hat{\boldsymbol{f}}(\boldsymbol{X}_B)) = \mathrm{HV}(\boldsymbol{y}_{t-1} \cup \hat{\boldsymbol{f}}(\boldsymbol{X}_B)) - \mathrm{HV}(\boldsymbol{y}_{t-1}), \tag{14}$$

where $\boldsymbol{X}_B = \{\boldsymbol{x}^{(b)}\}_{b=1}^B$ are selected solutions, and $\hat{\boldsymbol{f}}(\boldsymbol{X}_B)$ are the surrogate values. In this work, we mainly use the LCB (12) as the surrogate value for Bayesian optimization, and provide an ablation study of different surrogate values in Appendix F.9.

A better trade-off set will have a larger hypervolume, and the true Pareto set always has the largest one. We want to select a set of $\boldsymbol{X}_B$ such that their corresponding objective values $\hat{\boldsymbol{f}}(\boldsymbol{X}_B)$ maximize $\mathrm{HVI}(\hat{\boldsymbol{f}}(\boldsymbol{X}_B))$. It would be computationally expensive to jointly optimize a set of solutions to exactly maximize the hypervolume improvement (14), and therefore sequential greedy selection is typically used [14]. In this work, we select the set $\boldsymbol{X}_B$ in a sequential greedy manner from $\boldsymbol{X}$ where $|\boldsymbol{X}| = P = 1,000$ for all problems. More details can be found in Appendix D.2.

## 5 Experiments

In this section, we compare the proposed PSL method with other MOBO approaches on the performance of evaluated solutions. We also analyze the quality of the learned Pareto set model, which other methods cannot produce.

**Baseline Algorithms.** We consider several widely-used MOBO methods and two model-free approaches as baselines. The implementations of NSGA-II [20], MOEA/D-EGO [99], TSEMO [10], USeMO-EI [7], DGEMO [53] are from DGEMO's open-source codebase[1] based on pymoo[2] [8]. The implementations of scrambled Sobol sequence, qParEGO [45], TS-TCH [62], qEHVI [14] and qNEHVI [15] and from BoTorch[3] [5]. We implement the proposed PSL[4] in Pytorch [64].

**Benchmarks and Real-World Problems.** The algorithms are first compared on six newly proposed synthetic test instances (see Appendix E.1), as well as the widely-used VLMOP1-3 [88] and DTLZ2 [21] benchmark problems. Then we also conduct experiments on 5 different real-world multi-objective engineering design problems (RE) [85], including 1) four bar truss design [12]; 2) pressure vessel design [46]; 3) disk brake design [70]; 4) gear train design [19] and 5) rocket injector design [87]. Details of these problems can be found in Appendix E.

**Experiment Setting.** For each experiment, we randomly generate 10 initial solutions for expensive evaluations, and then conduct MOBO with 20 batched evaluations with batch size 5. Therefore, there are total 110 expensive evaluations. For an experiment, all algorithms are independently run 10 times. We use the hypervolume indicator [103] as the metric to compare the quality of evaluated solutions chosen by different MOBO algorithms, which is monotonic to the Pareto dominance relation. The ground truth Pareto front will always have the best (highest) hypervolume.

### 5.1 Experimental results and analysis

Table 1: Algorithm runtime per iteration (in seconds).

| Problem | #objs | MOEA/D-EGO | TSEMO | USeMO-EI | DGEMO | qEHVI | PSL(Ours): Model + Selection |
|---|---|---|---|---|---|---|---|
| F1 | 2 | 40.95 | 4.82 | 6.12 | 61.48 | 36.71 | 5.26 + 1.33 = 6.59 |
| DTLZ2 | 3 | 71.83 | 7.28 | 8.76 | 83.57 | 75.92 | 7.02 + 1.59 = 8.61 |

**MOBO Performance.** We compare PSL with other MOBO methods on the performance of evaluated solutions. Figure 5 shows the log hypervolume difference to the true/approximate Pareto front for the synthetic/real-world problems during the optimization process. The approximate Pareto fronts for the real-world design problems are from Tanabe and Ishibuchi [85] with a large number of evaluations, which are also used in other MOBO works. In most experiments, our proposed PSL method has better or comparable performance with other MOBO algorithms. Especially, as a generalized scalarization-based method, PSL significantly outperforms the model-free counterparts such as qParEGO [45, 15], MOEA/D-EGO [99], and TS-TCH [62]. These promising results validate the efficiency and usefulness of Pareto set learning for MOBO. More discussion of the proposed algorithm can be found in Appendix A.1 and Appendix A.2.

As shown in Table 1, PSL has a shorter or comparable total runtime (e.g., for modeling and batch selection) per iteration with other MOBO methods, which can be ignored in the expensive optimization problems (might take days). The algorithm runtimes for all problems can be found in Appendix F.1. These results confirm that the Pareto set learning approach has a low computational overhead which is affordable for MOBO.

**The Learned Pareto Set.** We present the approximate Pareto set learned by PSL under the posterior mean after optimization in Figure 6, which is not supported by other MOBO methods. According to the results, PSL can successfully learn the Pareto sets for different benchmarks and real-world application problems with different shapes of Pareto fronts. For benchmark problems, PSL can match the ground truth Pareto front with a small evaluation budget. For real-world applications, the approximate Pareto fronts can capture the trade-off among objectives and provide valuable information to support decision-making. We further discuss the the practicality of the approximate Pareto set in Appendix A.3.

---

[1] `https://github.com/yunshengtian/DGEMO`      [2] `https://pymoo.org/problems/index.html`
[3] `https://github.com/pytorch/botorch`    [4] `https://github.com/Xi-L/PSL-MOBO`

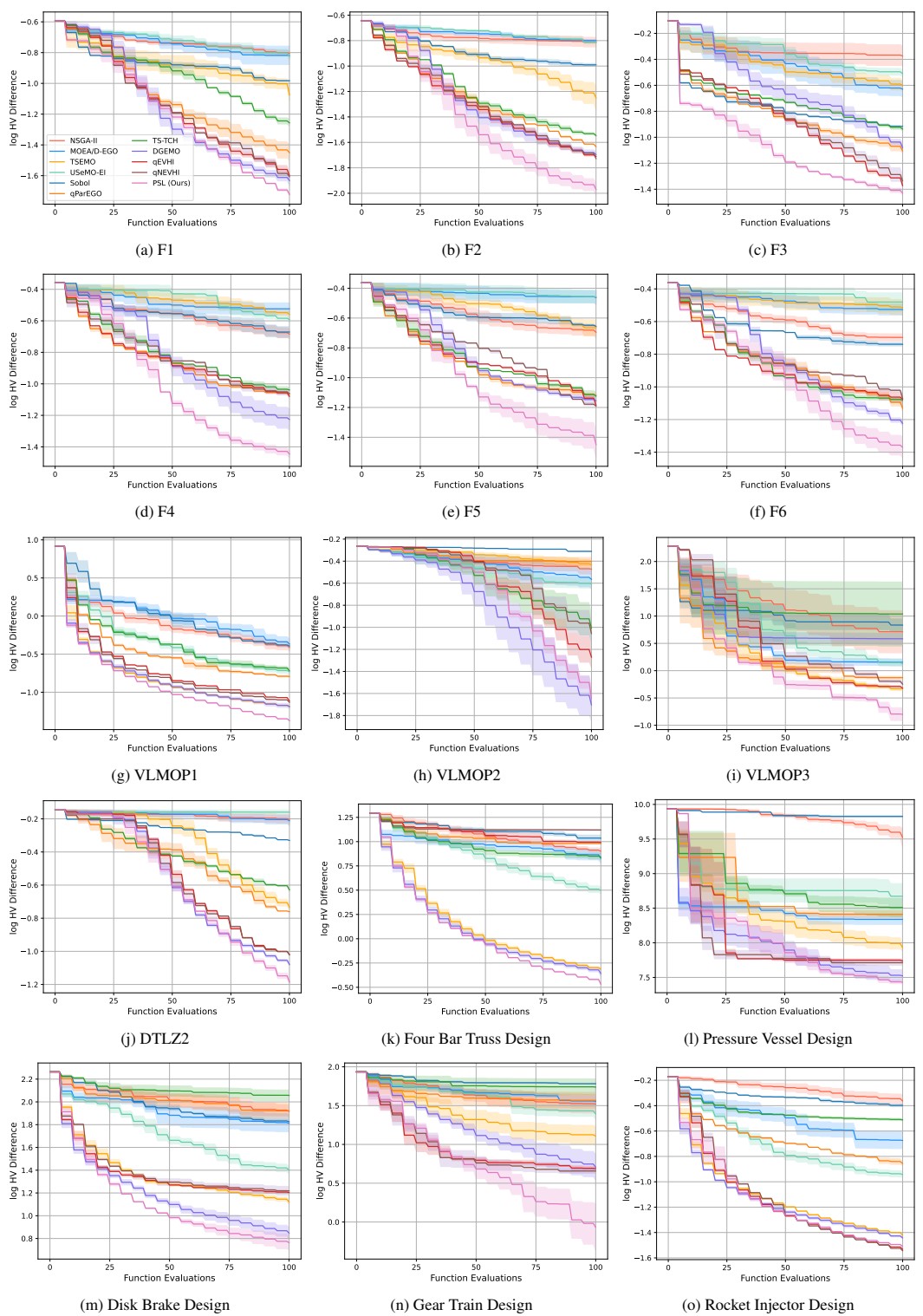

Figure 5: The log hypervolume difference w.r.t. the number of expensive evaluation of all algorithms for 15 different problems. The solid line is the mean value averaged over 10 independent runs for each algorithm, and the shaded region is the standard deviation around the mean value. **The labels of all algorithms can be found in Subfigure (a).**

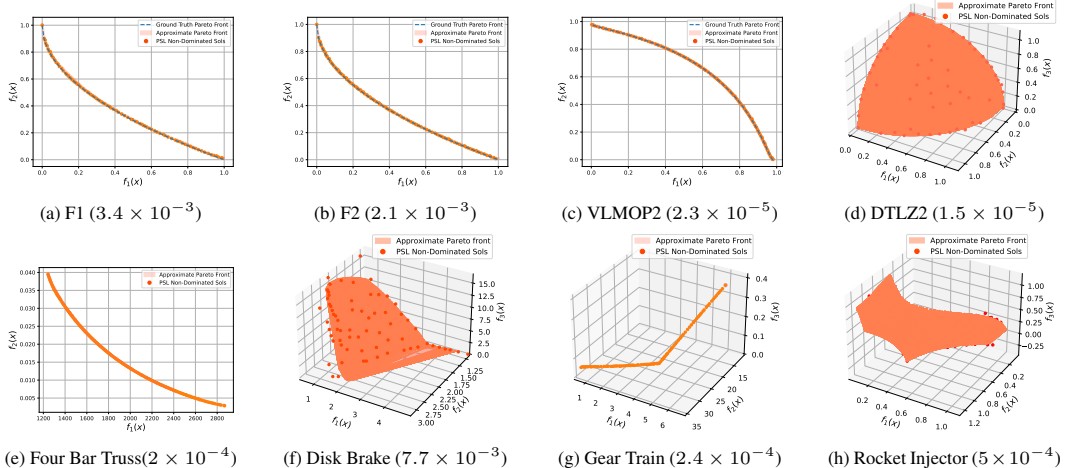

(a) F1 $(3.4 \times 10^{-3})$  (b) F2 $(2.1 \times 10^{-3})$  (c) VLMOP2 $(2.3 \times 10^{-5})$  (d) DTLZ2 $(1.5 \times 10^{-5})$

(e) Four Bar Truss $(2 \times 10^{-4})$  (f) Disk Brake $(7.7 \times 10^{-3})$  (g) Gear Train $(2.4 \times 10^{-4})$  (h) Rocket Injector $(5 \times 10^{-4})$

Figure 6: **The Learned Pareto Fronts (Relative Hypervolume Difference) by PSL:** Our learned Pareto fronts can match the ground truth Pareto fronts for the synthetic benchmarks, and have small relative hypervolume differences to the approximate Pareto fronts for real-world design problems. The learned Pareto front can well represent the optimal trade-offs among different objectives and provide valuable information to support flexible decision-making.

**Flexible Trade-off Adjustment.** With our model, the decision-makers can easily explore the whole approximate Pareto set by themselves to select the most preferred trade-off solution(s) as shown in Figure 7. No time-consuming communication between the optimization modeler and the decision-maker is required. By directly exploring the approximate Pareto front in an interactive manner, the decision-makers can observe and understand the connection between the trade-off preferences and corresponding solutions in real-time. It is also beneficial for decision-makers to further adjust and assign their most accurate preferences. The ability to incorporate user's knowledge into decision making [30] could be crucial for many real-world applications. More experimental results and analyses can be found in Appendix F.

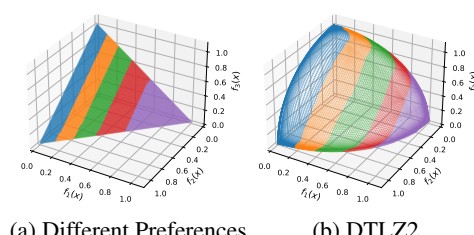

(a) Different Preferences  (b) DTLZ2

Figure 7: Different trade-off preferences and their corresponding solutions on the approximate Pareto set.

## 6 Conclusion, limitation and future work

**Conclusion.** We have proposed a novel Pareto set learning method, which is a first attempt to approximate the whole Pareto set for expensive multi-objective optimization. The advantages of this approach are two-fold. First, by learning and utilizing the approximate Pareto set, it can serve as an efficient MOBO method that outperforms different existing approaches. Secondly, it allows decision-makers to readily explore the whole approximate Pareto set, which supports flexible and interactive decision-making. We believe the proposed Pareto set learning method could provide a novel way for solve expensive multi-objective optimization.

**Limitation and Future Work.** The quality of the approximate Pareto set mainly depends on the accuracy of the surrogate models and the performance of the Pareto set learning algorithm, which could be poor for problems with insufficient evaluation budget and/or large-scale search space. A more detailed discussion of limitations and potential future work can be found in Appendix B, and potential societal impact can be found in Appendix C.

### Acknowledgements

This work was supported by the Hong Kong General Research Fund (11208121, CityU-9043148).

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
