We provide more discussion, details of the proposed algorithm and problem, and extra experimental results in this appendix:

- More discussions of the proposed algorithm are provided in Section A.
- The limitations and potential future improvement for PSL are discussed in Section B.
- Potential societal impact of this work is discussed in Section C.
- Details of the set model and batch selection algorithm are provided in Section D.
- Details of the benchmark and real-world application problems are given in Section E.
- More experimental results and analyses are presented in Section F.

# A    More discussion

## A.1    Motivation: Pareto set model over simple scalarization

Our proposed Pareto set learning method is closely related to the scalarization-based methods. But it can overcome a major disadvantage of the current scalarization methods for expensive optimization.

The scalarization methods randomly select one (e.g., ParEGO [45]) or a batch of scalarized subproblems (e.g., MOEA/D-EGO [99], TS-TCH [62], and qParEGO [15]) at each iteration. By optimizing the acquisition function (e.g., EI, LCB, Thompson Sampling) on each scalarization, they generate a batch of solutions for expensive evaluation.

A major limitation of all these scalarization methods is that they do not explicitly consider those already-evaluated solutions, neither for choosing the weighted scalarization, nor for selecting the next (batch of) solution(s) for expensive evaluation. Therefore, the selected weighted scalarization(s) might be close to those already-evaluated solutions. In other words, even the obtained solution(s) can maximize the EI/LCB for the selected scalarization, they could still be similar to those already-evaluated ones, and indeed not an optimal choice for multi-objective optimization. This limitation leads to inferior performance of these scalarization methods, as reported in [14, 53] and in our experimental results.

To overcome this limitation, our proposed method has a two-stage approach:

- Stage 1: Using the Pareto set model, it first efficiently samples a dense set of candidate solutions to cover the whole approximate Pareto front (of posterior mean, EI, or LCB etc).
- Stage 2: Then it selects a small batch of appropriate solutions from this dense set for expensive evaluation. For selection, we use the HVI criteria to take both the selected and already-evaluated solutions into consideration.

In this way, our proposed method can efficiently explore the whole approximate Pareto front, and choose the most appropriate solutions (on the approximate Pareto front, while far from the selected and already-evaluated ones) for expensive evaluation. The experimental results have validated the efficiency of the proposed method.

## A.2    Hypervolume improvement for batch selection

Efficiently finding a batch of solutions to optimize the EI/LCB of HVI could be very challenging. qEHVI [14] and qNEHVI [15] are two promising approaches along this direction. From the viewpoint of optimizing HVI, our proposed method restricts the search procedure only on the approximate Pareto set, which is a low-dimensional (e.g., $(m-1)$-dimensional) manifold in the decision space [36, 98]. Therefore, we can use a simple two-stage sample-then-select approach to find the batch of solutions for evaluations. For the optimal situation, the set of solutions that optimize HVI should all be on the Pareto front. However, if an efficient method exists, directly optimizing the EI/LCB of HVI on the whole search space should be a principled approach for batch selection.

On the other hand, the scalarization-based approach could have a close relationship with the hypervolume [100]. It will be very interesting to study how to better leverage this relation for designing a more efficient algorithm, such as learning the whole Pareto set while inference the location of solutions (on the learned Pareto set) to optimize HVI at a single stage. These will be our future work.

In summary, the advantages of PSL can be summarized from the following two viewpoints:

- From the viewpoint of scalarization based methods, PSL proposes a novel two-stage approach to select a small batch of appropriate solutions from the approximate Pareto front, which takes those already-evaluated solutions into consideration. As a result, it significantly outperforms other scalarization-based methods.

- From the viewpoint of hypervolume improvement based methods, in an ideal case, the set of solutions that optimize HVI should all be on the Pareto front. Restricting the search on the approximate Pareto set, PSL leads to an efficient HVI algorithm.

### A.3 Practicality of the approximate Pareto set

The learned approximate Pareto set could not always be accurate in real-world applications. Therefore, it is risky to only rely on the approximate Pareto set to make a decision (see more discussion in Section. B). With our proposed method, decision-makers can simultaneously have the evaluated solutions and the approximate Pareto set. The approximate Pareto set can provide extra useful information to better support their decision-making such as:

- It can help decision-makers to better understand the (approximate) trade-offs they will make for choosing any already evaluated solution. The approximate Pareto set and the corresponding surrogate objective values around the chosen solution can provide valuable information to understand what we can gain or lose by adjusting the chosen solution.

- It allows decision-makers to explore the whole approximate Pareto set easily. If none of the already evaluated solutions can satisfy the decision-maker's preferred trade-off, the decision-maker can rely on the approximate Pareto set/front to choose the preferred solution for further evaluation.

- When the optimization modeler and the final decision-maker are not the same person, the approximate Pareto set/front provides a much more efficient way for them to communicate and discuss the whole trade-offs among different objectives during or after the optimization process. This demand is common and essential for many applications [56].

- Finally, an important application for MOBO is to help domain experts efficiently conduct experiments. The approximate Pareto set might contain useful patterns and structures which can help the domain expert obtain more information from the experiment. It also provides a way for domain experts to incorporate their knowledge into the optimization process (e.g., choose the most concerned region, and eliminate some uninteresting locations). The proposed Pareto set learning method could be a novel approach to support the "bringing decision-makers back into the loop" approach for Bayesian optimization [30].

Therefore, the learned approximate Pareto set could be a useful tool to support flexible decision-making for expensive multi-objective optimization.

## B Limitation and potential future work

### B.1 The approximate Pareto set could be inaccurate

It could be very risky, especially in those safety-critical applications, to only rely on an approximate Pareto set to make the final decision. The quality of the approximate Pareto set heavily depends on the quality of the surrogate models. We cannot obtain a good approximate Pareto set if the evaluation budget is insufficient to build a good set of surrogate models. This is also a general challenge for (multi-objective) Bayesian optimization [30].

One possible method to address this challenge is to leverage the user preference information into the expensive optimization process [3, 4, 62]. In this way, the general MOBO method can spend the limited evaluation budget mainly on the user-preferred region rather than the whole decision space. Similarly, our proposed method can only build a partial approximate Pareto set with the user-preferred trade-offs. However, the preference-based approach could be affected by the following limitation.

### B.2 Defining the user preference is challenging for black-box optimization

Although some scalarization methods (e.g., Chebyshev scalarization) have good theoretical properties to connect the scalars to their corresponding Pareto solutions [13], it could still be hard to define user preferences in terms of scalars, especially in the black-box expensive optimization setting.

In many applications, the decision-makers might not even know their actual preferences before making their decision. Suppose the approximate Pareto set can be learned appropriately, instead of asking users to provide their preferences, our proposed approach can let them interactively explore the whole approximate Pareto set/front to select their most preferred solutions. In this interactive way, it could be much easier for the decision-makers to accurately express and assign their preferences. However, it could be difficult to precisely obtain user preference if a good approximate Pareto set is unavailable (e.g., in the early stage of optimization, or without enough budget).

The algorithm proposed in Abdolshah et al. [3] is an elegant and promising way to incorporate preference into the MOBO process via the preference-order constraints, which does not require any prior knowledge on the (approximate) Pareto front. The preference-order constraint also has a close relationship with the lexicographic approach for multi-objective optimization [72], which is a non-scalarizing method with good theoretical property (e.g., connection to weakly Pareto optimal solution). Incorporating this information into our proposed Pareto set learning model for more flexible preference incorporation could be interesting for future work.

### B.3 Scalability

The scalability of the search space dimension is a major challenge for general MOBO algorithms. It is also very difficult for our proposed PSL method to learn a good enough Pareto set for such large-scale problems. The PSL's performance depends on a set of good surrogate models, which requires a large number of evaluated solutions for the problem with a high-dimensional search space.

Recently, some efficient search region decomposition/management methods have been proposed to better allocate the limited computational budget for problems with high-dimensional search space [27, 89]. These ideas can be naturally generalized to solve expensive multi-objective optimization problems, such as MORBO [17] and LaMOO [101]. Since the search region management methods are algorithm-agnostic (e.g., a meta-algorithm), they can be combined with different (multi-objective) optimization algorithms. Studying how to efficiently combine the MORBO/LaMOO approach with our proposed PSL method will be an important future work to tackle this limitation.

## C  Potential societal impact

Our proposed Pareto set learning method mainly has two strengths: (1) it is good for better multi-objective Bayesian optimization in terms of speed and sample efficiency, and (2) it can help decision-makers to navigate the estimated Pareto set for better decision-making. These strengths could lead to many positive potential societal impacts. For example, as an efficient MOBO algorithm, it can reduce the cost of obtaining a set of diverse Pareto solutions in many applications, such as materials science, engineering design, and recommender systems. The learned approximate Pareto set provides a novel way for decision-makers to easily explore different trade-offs, which could be an important component for a more user-friendly multi-criteria decision system.

On the negative side, as discussed in the limitation section, there is no guarantee that the learned approximate Pareto set could be accurate. Solely relying on the approximate Pareto set to make a decision could be risky. The decision-makers should leverage all the information they have to make the final decision. In addition, the leakage of the learned Pareto set model might unintentionally reveal the problem information and user preference, which should be avoided in real-world applications.

# D   Model and algorithm details

## D.1   Pareto set model

**Model Structure.** In this work, we use a multi-layered perceptron (MLP) as the Pareto set model $x(\lambda) = h_\theta(\lambda)$. For all experiments, the Pareto set model has 2 hidden layers each with 256 hidden units, and the activation is ReLU. The input and output dimension of the set model is the number of objectives $m$ and the dimension of decision variables $n$, respectively. Therefore, the set model $h_\theta(\lambda)$ has the following structure:

$$
\begin{aligned}
h_\theta(\lambda) : &\text{Input } \lambda \to \text{Linear}(m, 256) \to \text{ReLU} \to \text{Linear}(256, 256) \\
&\to \text{ReLU} \to \text{Linear}(256, 256) \to \text{ReLU} \\
&\to \text{Linear}(256, n) \to \text{Output } x(\lambda),
\end{aligned}
\tag{15}
$$

where the input is an $m$-dimensional preference $\lambda \in \Lambda = \{\lambda \in \mathbb{R}_+^m | \sum \lambda_i = 1\}$, the output is a $n$-dimensional decision variables $x \in \mathbb{R}^n$, and $\theta \in \mathbb{R}^d$ represents all learnable parameters for the Pareto set model. We build independent Gaussian process models with Matérn 5/2 kernel for each objective as the surrogate models. The objective to optimize is the augmented Tchebycheff scalarization on the surrogate value with respect to all valid preferences.

**Model Training.** In the proposed Pareto set learning method, we have the following model structure:

$$
\lambda \to \textbf{Pareto Set Model} \to x(\lambda) = h_\theta(\lambda)
\tag{16}
$$

$$
\to \textbf{GP Model} \to \hat{f}(x) \to \textbf{TCH Scalarization} \to \hat{g}_{\text{tch\_aug}}(x(\lambda)),
\tag{17}
$$

where the preference $\lambda$ goes through the Pareto set model $h_\theta(\lambda)$ and Gaussian process model, then gets the augmented Tchebycheff scalarized value $\hat{g}_{\text{tch\_aug}}(x(\lambda))$ at the end (a detailed version can be found in Figure 4 in the main paper). To train the Pareto set model, we use a gradient-based method to optimize the model parameter $\theta$ with respect to the scalarized value $\hat{g}_{\text{tch\_aug}}(x(\lambda))$. With the chain rule, we have:

$$
\nabla_\theta \hat{g}_{\text{tch\_aug}}(x = h_\theta(\lambda) | \lambda) = \frac{\partial \hat{g}_{\text{tch\_aug}}}{\partial \theta} = \frac{\partial \hat{g}_{\text{tch\_aug}}}{\partial \hat{f}} \frac{\partial \hat{f}}{\partial x} \frac{\partial x}{\partial \theta},
\tag{18}
$$

where we can calculate each term respectively.

- For the first term $\frac{\partial \hat{g}_{\text{tch\_aug}}}{\partial \hat{f}}$, the augmented Tchebycheff scalarization $\hat{g}_{\text{tch\_aug}}(x | \lambda) = \max_{1 \le i \le m}\{\lambda_i(\hat{f}_i(x) - (z_i^* - \varepsilon))\} + \rho \sum_{i=1}^m \lambda_i \hat{f}_i(x)$ has a max operator which is technically only subdifferentiable, but it is known to have good subgradients [94] for surrogate model based optimization. We simply take the subgradients for this term.

- The second term $\frac{\partial \hat{f}}{\partial x}$ is the gradient of surrogate values to the input $x$. In this work, we build independent Gaussian process models for each objective with Matérn 5/2 kernel. The gradients of the kernel $\frac{\partial k_i}{\partial x}$, predictive mean $\frac{\partial \hat{\mu}_i}{\partial x}$ and standard deviation $\frac{\partial \hat{\sigma}_i}{\partial x}$ for each objective can be easily calculated. With these terms, the calculation for the surrogate value (e.g., predictive mean, UCB and EI) for each objective $\frac{\partial \hat{f}_i}{\partial x}$ is also straightforward.

- The third term $\frac{\partial x}{\partial \theta} = \frac{\partial h_\theta(x)}{\partial \theta}$ is the gradient of the Pareto set model to the parameter $\theta$. In this work, we use a simple MLP as the Pareto set model, and its gradient be easily obtained by auto-differentiation such as in PyTorch [64].

For **Algorithm 1** in the main paper, we randomly sample $K = 10$ different preferences $\{\lambda_k\}_{k=1}^{K=10} \sim \Lambda$ in batch at each step. Without any prior information (e.g., user's preference), we uniformly sample preference $\lambda$ from $[0, 1]^m$ and then normalize it such that $\sum_{i=1}^m \lambda_i = 1$. The total update step is $T = 1000$ for training the Pareto set model at each iteration. The model optimizer is Adam with learning rate $\eta = 1e - 3$ and no weight decay. The learning process typically only requires less than 10 seconds to finish, which is enough to obtain a good Pareto set approximation. Detailed experimental results on the runtime can be found in Table 3.

We use the same model and training procedure for all experiments, and the only problem-dependent setting is the input/output dimension $m$ and $n$. The proposed Pareto set model and its learning approach have robust and promising performances for different expensive multi-objective optimization benchmarks and real-world problems.

## D.2 Batch selection with Pareto set learning

---
**Algorithm 4** Sampling-based Greedy Batch Selection
---
1: **Input:** Evaluated Solutions $\{\boldsymbol{X}_{t-1}, \boldsymbol{y}_{t-1}\}$, Candidate Solutions $\boldsymbol{X}_C = \boldsymbol{X}$, Surrogate Models
2: Initialize the selected solution set $\boldsymbol{X}_B = \varnothing$ with predicted values $\hat{\boldsymbol{y}}_B = \varnothing$
3: **for** $t = 1$ to $B$ **do**
4:     Calculate predicted HVIs for each individual solution in $\boldsymbol{X}_C$ with respective to $\boldsymbol{y}_{t-1} \cup \hat{\boldsymbol{y}}_B$
5:     Choose the single solution $\boldsymbol{x}_s$ with the best predicted HVI
6:     $\boldsymbol{X}_B \leftarrow \boldsymbol{X}_B \cup \{\boldsymbol{x}_s\}, \hat{\boldsymbol{y}}_B \leftarrow \hat{\boldsymbol{y}}_B \cup \{\hat{\boldsymbol{f}}(\boldsymbol{x}_s)\}, \boldsymbol{X}_C \leftarrow \boldsymbol{X}_C \setminus \{\boldsymbol{x}_s\}$
7: **end for**
8: **Output:** The Selected Solutions $\boldsymbol{X}_B$.
---

The MOBO with Pareto Set Learning framework (e.g., **Algorithm 2**) is similar to other MOBO methods, and the main difference is on the batch selection with the learned Pareto set (**Algorithm 3**). At each iteration, we first randomly sample $P = 1000$ preferences $\{\boldsymbol{\lambda}^{(p)}\}_{p=1}^{P=1000} \sim \Lambda$ and directly obtain their corresponding solutions $\boldsymbol{X} = \{\boldsymbol{x}(\boldsymbol{\lambda}^{(p)})\}_{p=1}^{P=1000}$ on the current learned Pareto set $\mathcal{M}_{\text{psl}}$. We then use the predicted hypervolume improvement (HVI) with respect to the already evaluated solutions $\{\boldsymbol{X}_{t-1}, \boldsymbol{y}_{t-1}\}$ to select a subset $\boldsymbol{X}_B \subset \boldsymbol{X}$ for expensive evaluation. The surrogate values for the candidate solutions are directly used for the predicted HVI calculation if they are the predicted mean $\hat{\boldsymbol{\mu}}$ or the lower confidence bound $\hat{\boldsymbol{\mu}} - \beta\boldsymbol{\sigma}$. When the surrogate values are expected improvement, although the approximate Pareto set is learned with EI for each scalarization, we let $\hat{\boldsymbol{f}}(\boldsymbol{x}_s) = \hat{\boldsymbol{\mu}} - \boldsymbol{\sigma}$ for the predicted HVI calculation. The main reasons for this choice are: 1) we want to avoid the (repeatedly) time-consuming Monte Carlo integration for calculating the expected hypervolume improvement; 2) the LCB (or the posterior mean only) is on the same scale as the value of those already-evaluated solutions, which make the calculation of HVI meaningful.

Since it is computationally intensive to find $B$ solutions to exactly optimize HVI, we select the solution batch $\boldsymbol{X}_B$ in a sequentially greedy manner from $\boldsymbol{X}$. Similar approaches have also been used in the current MOBO methods [53, 17]. The greedy selection approach is presented in **Algorithm 4**. We sequentially sample solutions from the candidate set $\boldsymbol{X}_C$, and add the single solution $\boldsymbol{x}_s$ with the best HVI value into the selected set $\boldsymbol{X}_B$. For all problems, the acquisition search procedure (solution sampling + individual search + batch selection) with batch size $B = 5$ typically costs less than 3 seconds. The detailed results can be found in Table 3.

# E  Problem details

Table 2: Problem information and reference point for both synthetic benchmarks and real-world engineering design problems.

| Problem | n | m | Reference Point($r$) |
|---|---|---|---|
| F1 | 6 | 2 | (1.1,1.1) |
| F2 | 6 | 2 | (1.1,1.1) |
| F3 | 6 | 2 | (1.1,1.1) |
| F4 | 6 | 2 | (1.1,1.1) |
| F5 | 6 | 2 | (1.1,1.1) |
| F6 | 6 | 2 | (1.1,1.1) |
| VLMOP1 | 1 | 2 | (4.4,4.4) |
| VLMOP2 | 6 | 2 | (1.1,1.1) |
| VLMOP3 | 2 | 3 | (11,66,1.1) |
| DTLZ2 | 6 | 3 | (1.1,1.1,1.1) |
| Four Bar Truss Design | 4 | 2 | (3175.0065, 0.0400) |
| Pressure Vessel Design | 4 | 2 | (6437.2649, 1417536.7586) |
| Disk Brake Design | 4 | 3 | (5.8374, 3.4412, 27.5) |
| Gear Train Design | 4 | 3 | (6.5241, 61.6, 0.3913) |
| Rocket Injector Design | 4 | 3 | (1.0884, 1.0522, 1.0863) |

## E.1  Synthetic benchmark problems

To better evaluate our proposed PSL method, we propose 6 new synthetic test problems with different shapes of Pareto sets which can be found in next page. We also test our proposed PSL algorithm on different widely-used synthetic multi-objective optimization benchmark problems, namely VLMOP1-3 [88] and DTLZ2 [21]. The input and output dimensions of these problems are shown in Table 2. These synthetic problems have known Pareto sets $\mathcal{M}_{\mathrm{ps}}$ and Pareto fronts $\boldsymbol{f}(\mathcal{M}_{\mathrm{ps}})$ with the nadir point:

$$\boldsymbol{z}_{\mathrm{nadir}} = (\max f_1(\boldsymbol{x}_1), \ldots, \max f_m(\boldsymbol{x}_m)), \tag{19}$$
$$\forall \boldsymbol{x}_1, \boldsymbol{x}_2, \ldots, \boldsymbol{x}_m \in \mathcal{M}_{\mathrm{ps}},$$

where $\boldsymbol{x}_1, \ldots, \boldsymbol{x}_m$ are solutions in the Pareto set. We set the reference point $\boldsymbol{r} = 1.1 \times \boldsymbol{z}_{\mathrm{nadir}}$ for each problem.

| | |
|---|---|
| F1 | $f_1(\boldsymbol{x}) = (1 + \frac{s_1}{|J_1|})\boldsymbol{x}_1, \quad f_2(\boldsymbol{x}) = (1 + \frac{s_2}{|J_2|})\left(1 - \sqrt{\frac{\boldsymbol{x}_1}{1 + \frac{s_2}{|J_2|}}}\right)$ 

 where $s_1 = \sum_{j \in J_1}(\boldsymbol{x}_j - (2\boldsymbol{x}_1 - 1)^2)^2$ and $s_2 = \sum_{j \in J_2}(\boldsymbol{x}_j - (2\boldsymbol{x}_1 - 1)^2)^2$, 

 $J_1 = \{j | j \text{ is odd and } 2 \le j \le n\}$ and $J_2 = \{j | j \text{ is even and } 2 \le j \le n\}$. |
| F2 | $f_1(\boldsymbol{x}) = (1 + \frac{s_1}{|J_1|})\boldsymbol{x}_1, \quad f_2(\boldsymbol{x}) = (1 + \frac{s_2}{|J_2|})\left(1 - \sqrt{\frac{\boldsymbol{x}_1}{1 + \frac{s_2}{|J_2|}}}\right)$ 

 where $s_1 = \sum_{j \in J_1}(\boldsymbol{x}_j - \boldsymbol{x}_1^{0.5(1.0 + \frac{3(j-2)}{n-2})})^2$ and $s_2 = \sum_{j \in J_2}(\boldsymbol{x}_j - \boldsymbol{x}_1^{0.5(1.0 + \frac{3(j-2)}{n-2})})^2$, 

 $J_1 = \{j | j \text{ is odd and } 2 \le j \le n\}$ and $J_2 = \{j | j \text{ is even and } 2 \le j \le n\}$. |
| F3 | $f_1(\boldsymbol{x}) = (1 + \frac{s_1}{|J_1|})\boldsymbol{x}_1, \quad f_2(\boldsymbol{x}) = (1 + \frac{s_2}{|J_2|})\left(1 - \sqrt{\frac{\boldsymbol{x}_1}{1 + \frac{s_2}{|J_2|}}}\right)$ 

 where $s_1 = \sum_{j \in J_1}(\boldsymbol{x}_j - \sin(4\pi\boldsymbol{x}_1 + \frac{j\pi}{n}))^2$ and $s_2 = \sum_{j \in J_2}(\boldsymbol{x}_j - \sin(4\pi\boldsymbol{x}_1 + \frac{j\pi}{n}))^2$, 

 $J_1 = \{j | j \text{ is odd and } 2 \le j \le n\}$ and $J_2 = \{j | j \text{ is even and } 2 \le j \le n\}$. |
| F4 | $f_1(\boldsymbol{x}) = (1 + \frac{s_1}{|J_1|})\boldsymbol{x}_1, \quad f_2(\boldsymbol{x}) = (1 + \frac{s_2}{|J_2|})\left(1 - \sqrt{\frac{\boldsymbol{x}_1}{1 + \frac{s_2}{|J_2|}}}\right)$ 

 where $s_1 = \sum_{j \in J_1}(\boldsymbol{x}_j - 0.8\boldsymbol{x}_1\cos(4\pi\boldsymbol{x}_1 + \frac{j\pi}{n}))^2$ and $s_2 = \sum_{j \in J_2}(\boldsymbol{x}_j - 0.8\boldsymbol{x}_1\sin(4\pi\boldsymbol{x}_1 + \frac{j\pi}{n}))^2$, 

 $J_1 = \{j | j \text{ is odd and } 2 \le j \le n\}$ and $J_2 = \{j | j \text{ is even and } 2 \le j \le n\}$. |
| F5 | $f_1(\boldsymbol{x}) = (1 + \frac{s_1}{|J_1|})\boldsymbol{x}_1, \quad f_2(\boldsymbol{x}) = (1 + \frac{s_2}{|J_2|})\left(1 - \sqrt{\frac{\boldsymbol{x}_1}{1 + \frac{s_2}{|J_2|}}}\right)$, 

 where $s_1 = \sum_{j \in J_1}(\boldsymbol{x}_j - 0.8\boldsymbol{x}_1\cos(\frac{4\pi\boldsymbol{x}_1 + \frac{j\pi}{n}}{3}))^2$ and $s_2 = \sum_{j \in J_2}(\boldsymbol{x}_j - 0.8\boldsymbol{x}_1\sin(4\pi\boldsymbol{x}_1 + \frac{j\pi}{n}))^2$, 

 $J_1 = \{j | j \text{ is odd and } 2 \le j \le n\}$ and $J_2 = \{j | j \text{ is even and } 2 \le j \le n\}$. |
| F6 | $f_1(\boldsymbol{x}) = (1 + \frac{s_1}{|J_1|})\boldsymbol{x}_1, \quad f_2(\boldsymbol{x}) = (1 + \frac{s_2}{|J_2|})\left(1 - \sqrt{\frac{\boldsymbol{x}_1}{1 + \frac{s_2}{|J_2|}}}\right)$ 

 where $s_1 = \sum_{j \in J_1}\{\boldsymbol{x}_j - [0.3\boldsymbol{x}_1^2\cos(12\pi\boldsymbol{x}_1 + \frac{4j\pi}{n})) + 0.6\boldsymbol{x}_1]\cos(6\pi\boldsymbol{x}_1 + \frac{j\pi}{n})\}$ 

 and $s_2 = \sum_{j \in J_2}\{\boldsymbol{x}_j - [0.3\boldsymbol{x}_1^2\cos(12\pi\boldsymbol{x}_1 + \frac{4j\pi}{n})) + 0.6\boldsymbol{x}_1]\sin(6\pi\boldsymbol{x}_1 + \frac{j\pi}{n})\}$, 

 $J_1 = \{j | j \text{ is odd and } 2 \le j \le n\}$ and $J_2 = \{j | j \text{ is even and } 2 \le j \le n\}$. |

## E.2 Real-world application problems

We also conduct experiments on 5 real-world multi-objective engineering design problems [85] that are initially proposed in different communities for different applications:

**Four Bar Truss Design.** This problem is to design a four-bar truss. The two objectives to optimize are its structural volume and joint displacement. The decision variables are the length of the four bars. The details of this problem can be found in Cheng and Li [12].

**Pressure Vessel Design.** This problem is to design a cylindrical pressure vessel. The two objectives to minimize are the total cost (material, forming, and welding) and the violations of three different design constraints. This problem has four decision variables, which are the shell thicknesses, the pressure vessel head, the inner radius, and the length of the cylindrical section. The details of this problem can be found in Kramer [46].

**Disk Brake Design.** This problem is to design a disc brake. The three objectives to minimize are the mass, the minimum stopping time, and the violations of four design constraints. This problem has four decision variables: the inner radius, the outer radius, the engaging force, and the number of friction surfaces. The details of this problem can be found in Ray and Liew [70].

**Gear Train Design.** This problem is to design a gear train with four gears. The three objectives to minimize are the difference between the realized gear ration and the required specification, the maximum size of four gears, and the design constraint violations. The decision variables are the numbers of teeth in each of the four gears. The details of this problem can be found in Deb and Srinivasan [19].

**Rocket Injector Design.** This problem is to design a rocket injector that needs to minimize the maximum temperature of the injector face, the distance from the inlet, and the temperature on the post tip. It has four decision variables, namely, the hydrogen flow angle, the hydrogen area, the oxygen area, and the oxidizer post tip thickness. The details of this problem can be found in Vaidyanathan et al. [87].

These real-world multi-objective design problems do not have known exact Pareto fronts. We use the approximate Pareto fronts provided by Tanabe and Ishibuchi [85] with a large number of evaluations as our refereed Pareto fronts, which have also been used in other MOBO works. We set the reference point $r = 1.1 \times \hat{z}_{\text{nadir}}$ where $\hat{z}_{\text{nadir}}$ is the nadir point of the approximate Pareto front. The problem information and reference points can be found in Table 2.

## E.3 Experiment settings

For all experiments, we first randomly sample and evaluate 10 valid solutions with Latin hypercube sampling [57], and set them as the initial solutions $\{X_0, y_0\}$. Then we further run the MOBO algorithm to optimize the given problem with a limited evaluation budget (100 for all problems). In the main paper, we report the results with a batch size 5, and there are only $100/5 = 20$ batched iterations for each algorithm. In section F in this appendix, we also report experimental results with different batch sizes.

# F  Additional experiments

## F.1  Run time

Table 3: Algorithm run time per iteration (in seconds).

| Problem | MOEA/D-EGO | TSEMO | USeMO-EI | DGEMO | qEHVI | PSL(Ours): Model + Selection |
|---|---|---|---|---|---|---|
| F1 | 55.21 | 4.82 | 6.12 | 61.48 | 36.71 | 5.26 + 1.33 = 6.59 |
| F2 | 60.12 | 5.18 | 7.06 | 62.21 | 42.80 | 6.01 + 1.21 = 7.22 |
| F3 | 58.66 | 5.63 | 6.84 | 59.88 | 38.28 | 5.82 + 1.43 = 7.25 |
| F4 | 57.81 | 5.03 | 6.51 | 61.73 | 40.16 | 5.72 + 1.15 = 6.87 |
| F5 | 63.08 | 5.41 | 7.24 | 57.29 | 38.83 | 5.53 + 1.38 = 6.91 |
| F6 | 58.57 | 5.26 | 5.97 | 69.71 | 42.77 | 5.89 + 1.19 = 7.08 |
| VLMOP1 | 57.13 | 4.19 | 5.23 | 60.33 | 28.37 | 3.88 + 1.06 = 4.94 |
| VLMOP2 | 61.05 | 4.96 | 6.88 | 68.20 | 39.51 | 4.62 + 1.29 = 5.91 |
| VLMOP3 | 68.72 | 8.20 | 9.03 | 79.82 | 71.25 | 5.61 + 2.49 = 8.10 |
| DTLZ2 | 71.83 | 7.28 | 8.76 | 83.57 | 75.92 | 7.02 + 1.59 = 8.61 |
| Four Bar Truss | 63.29 | 3.83 | 5.92 | 62.46 | 42.73 | 6.01 + 0.78 = 6.79 |
| Pressure Vessel | 62.61 | 4.61 | 6.59 | 69.73 | 45.32 | 4.98 + 1.33 = 6.31 |
| Disk Brake | 72.54 | 7.03 | 8.62 | 75.20 | 88.79 | 5.50 + 2.28 = 7.78 |
| Gear Train | 68.48 | 6.92 | 9.31 | 84.52 | 79.55 | 4.24 + 2.13 = 6.37 |
| Rocket Injector | 69.43 | 8.72 | 10.03 | 79.31 | 85.30 | 5.09 + 2.26 = 7.35 |

We report the run time for each algorithm with batch size 10 per iteration in Table 3. For our proposed PSL algorithm, we further report the detailed run time for learning the set model and for searching the batch solutions. According to the results, PSL can efficiently learn the set model and select a batch of solutions for evaluation at each iteration with a total time budget of less than 10 seconds. The short PSL batch selection run time is expected, since all solutions are directly sampled from the learned Pareto set with a few further local search steps, rather than optimizing the acquisition function(s) from scratch as in other MOBO methods.

We want to emphasize that the run time for different MOBO algorithms strongly depends on the implementation, which is also highlighted in the current works [14, 53]. The objective evaluations in real-world applications are usually very expensive and involve costly real-world experiments or simulations that take days to run. All the MOBO algorithms are efficient and have negligible run-time overheads in these real-world application scenarios.

## F.2 Quality of the learned Pareto set

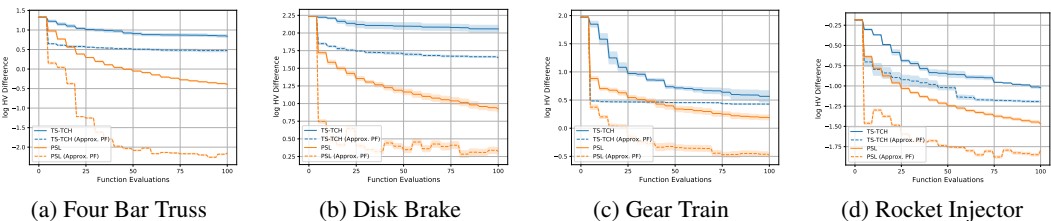

| (a) Four Bar Truss | (b) Disk Brake | (c) Gear Train | (d) Rocket Injector |

Figure 8: The log hypervolume difference w.r.t. the number of expensive evaluations for PSL and TS-TCH. We report both results for the evaluated solutions (solid lines) and the approximate Pareto front (dashed line).

In the main paper, we have demonstrated that the proposed PSL method can successfully approximate the ground truth Pareto front with a limited evaluation budget as in Figure 6. In this subsection, we further investigate the performance of the learned Pareto fronts during the optimization process. As shown in Figure 8, the learned Pareto fronts have good quality (with 1,000 random samples) during optimization and lead to promising evaluated solutions performance.

In addition, we also compare our Pareto set model with another modeling approach based on evaluated solutions. Specifically, we run the TS-TCH algorithm and record the evaluated solutions with their corresponding preferences during the optimization process. Then we build a model to approximate the Pareto front based on the evaluated and non-dominated solutions at each iteration step. The model we built has an identical structure (2-layer MLP) to the PSL model, but is now trained in a supervised manner. According to the results in Figure 8, our proposed PSL model has better quality than the model trained in a supervised manner with evaluated solutions.

## F.3 Impact of the Pareto set model for scalarization-based method

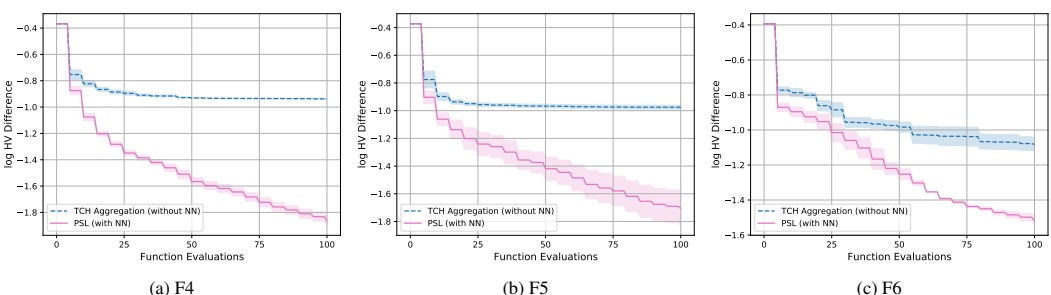

| (a) F4 | (b) F5 | (c) F6 |

Figure 9: The log hypervolume difference w.r.t. the number of expensive evaluation for different algorithms for F4, F5 and F6.

In this work, we choose the weighted Chebyshev scalarization with an ideal point and epsilon mainly due to its good theoretical property. In this subsection, we conduct an ablation study on our proposed PSL method with and without the Pareto set model. The results in Figure 9 show that simply optimizing the acquisition function for scalarization could lead to a significant performance drop. It confirms that the proposed Pareto set model is important for the overall promising performance.

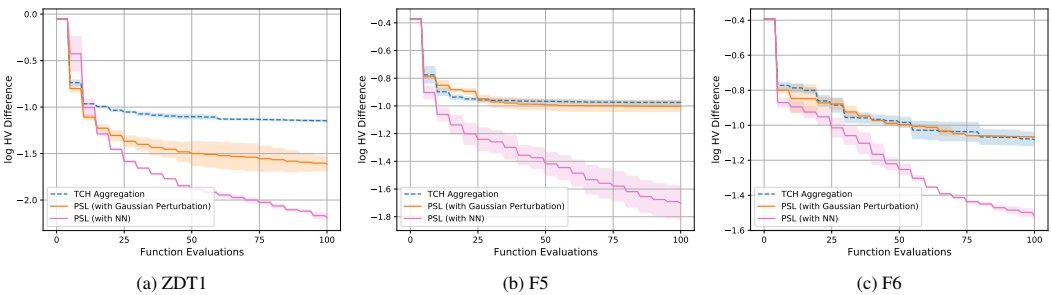

Figure 10: The HVI-LCB acquisition values obtained by different search methods along the PSL's optimization process on ZDT1, F5, and F6.

In this subsection, we conduct an ablation on small perturbing the best points v.s. Pareto set model for generating the candidate set. Based on the results shown in Figure 10, the Gaussian perturbing method can improve the performance of simple scalarization on problems with simple Pareto set (e.g., ZDT1 [102]) but not the problems with complicated Pareto set (e.g., F5 and F6). On all problems, our proposed PSL method still achieves the best performance.

DGEMO [53] also has a local search approach to expand the candidate set around the best points (in terms of surrogate value) with the first and second derivatives of the GP surrogate model. In our experiments, PSL can outperform DGEMO on most problems, which confirms the importance of the Pareto set model for generating the candidate set.

On the other hand, the local search methods might provide complementary candidate solutions to PSL, especially at the early stage of optimization when the approximate Pareto set is inaccurate. We will investigate how to efficiently combine PSL with the local search approaches in future work.

## F.5 Comparisons with HVI-LCB

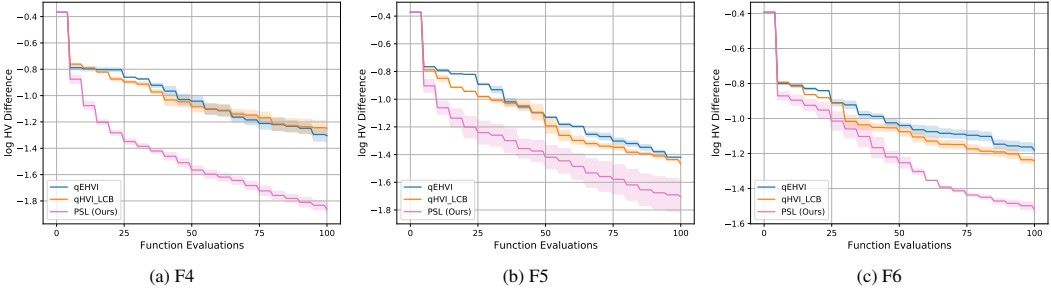

Figure 11: The log hypervolume difference w.r.t. the number of expensive evaluation for qEHVI, qHVI-LCB, and our proposed PSL on F4, F5, and F6.

In this subsection, we conduct an ablation study on maximizing the hypervolume improvement with LCB on the approximate Pareto set vs. searching across the whole design space. The experimental results on three newly proposed problems (i.e., F4, F5, and F6) with complicated Pareto sets are shown in Figure 11. We also provide the qEHVI results as reference.

Based on these results, it is clear that our proposed PSL method can outperform directly searching the entire search space to maximize HVI with LCB. The qHVI-LCB method performs similarly to qEHVI, which indicates that different acquisitions (for HVI) do not have a significant impact on the optimization performance. In other words, with the learned Pareto set, our proposed PSL method can efficiently conduct the acquisition optimization mainly on the promising low-dimensional (e.g., (m-1)-dimensional) manifold, and then lead to significantly better Bayesian optimization performance.

One possible reason for this performance gap could be the difficulty of directly searching the whole design search for maximizing HVI. As shown in recent works [101, 17], the performance of qEHVI

can be improved using meta search region management algorithms such as LaMOO [101] and trust region methods [17]. It implies that directly searching the whole design space is not efficient. These meta-algorithms adaptively decompose the whole search region into different subregions along with the optimization process, while our PSL method directly learns the whole approximate Pareto set at each step. Studying how to efficiently combine PSL with these algorithms could be an important future work as discussed in Appendix B.

## F.6    The HVI acquisition values during optimization

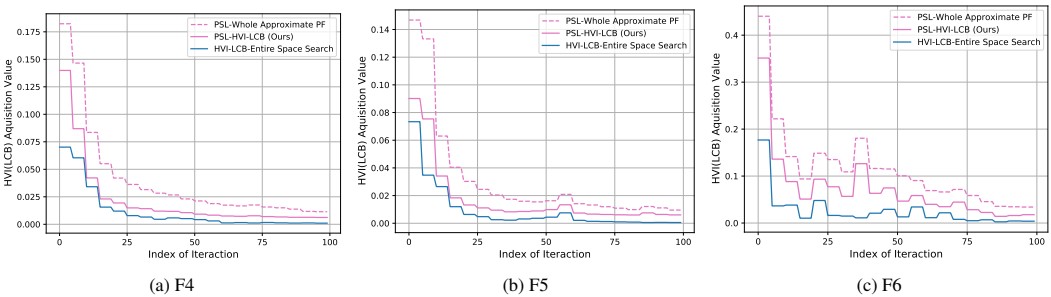

(a) F4                          (b) F5                          (c) F6

Figure 12: The HVI-LCB acquisition values obtained by different search methods along the PSL's optimization process on F4, F5, and F6.

In this subsection, we compare the HVI-LCB acquisition values obtained by different search methods. We run our proposed PSL as the optimization algorithm, and conduct different search methods at each iteration. The compared search methods are: 1) searching across the entire design space, 2) our PLS search on the approximate Pareto front, and 3) the HVI-LCB acquisition values for the whole approximate Pareto front with 1,000 sampled solutions (which can be treated as the upper bound for our method). Based on the results, it is clear that our proposed PSL search can produce better HVI-LCB acquisition values than directly searching the entire design space.

It should be noticed that the reported results are obtained by running PSL, and the search methods have the same surrogate functions and baseline HV (from the same evaluated solutions chosen by PSL) at each iteration. By running qHVI-LCB alone, the less efficient acquisition search at each iteration will lead to the accumulated worse overall performance as shown in Figure 11.

## F.7 PSL with different preferences

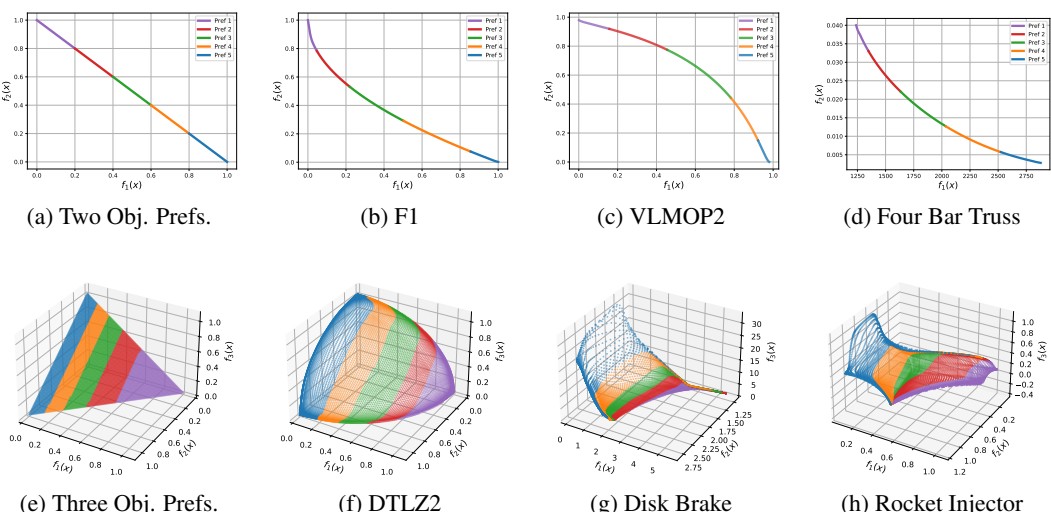

| (a) Two Obj. Prefs. | (b) F1 | (c) VLMOP2 | (d) Four Bar Truss |
| --- | --- | --- | --- |
| (e) Three Obj. Prefs. | (f) DTLZ2 | (g) Disk Brake | (h) Rocket Injector |

Figure 13: **Learned Pareto Set with Different Preferences:** Our proposed PSL model allows the decision-makers to flexibly explore the approximate Pareto front to obtain any trade-off solution. Suppose the decision-makers have different sets of preferred trade-offs, our model can successfully generate the corresponding parts of the Pareto fronts. The preference assignment is not fixed, and the decision-makers can adjust it and obtain the corresponding approximate solutions in real time.

Our proposed PSL method enables the decision-makers to easily adjust their preference to explore the whole approximate Pareto set for making decisions. This ability is not supported by other MOBO methods, which is important to flexibly incorporate the decision-maker's preference and prior knowledge into Bayesian optimization [30].

The decision-making process for real-world applications could be subjective. We follow a similar way with other preference-based MOBO to show our method can successfully generate different trade-off solutions. In Figure 13, suppose the decision-makers have different sets of trade-off preferences, our model can generate the corresponding parts of the Pareto fronts for different problems. We uniformly sample $1,000$ and $10,011$ preferences for the two-objective and three-objective problems respectively, and use the learned Pareto set models to generate the corresponding approximate Pareto solutions. We further divide the preferences into 5 exclusive parts, and label them with the corresponding solutions in different colors. As shown in Figure 13, the corresponding connection from the preferences to the solutions is clear for all problems.

By directly exploring the approximate Pareto front in an interactive manner, the decision-makers can observe and understand the connection between preferences and corresponding solutions in real-time. This ability could be beneficial for the decision-makers to further adjust and assign their most accurate preferences.

**F.8 Standard deviation on the approximated Pareto set**

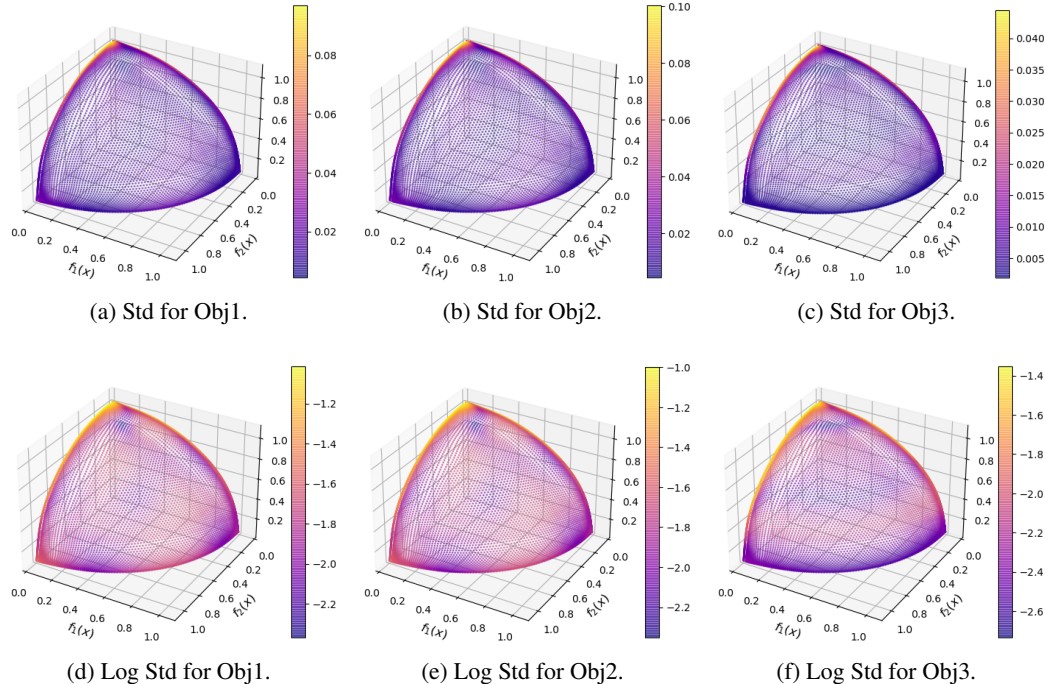

(a) Std for Obj1.      (b) Std for Obj2.      (c) Std for Obj3.

(d) Log Std for Obj1.      (e) Log Std for Obj2.      (f) Log Std for Obj3.

Figure 14: **Standard deviation and log standard deviation for DTLZ2:** The predicted standard deviations for different objectives on the approximated Pareto front.

In addition to the predicted mean, our PSL method also provides the uncertainty information (e.g., predicted standard deviation) to support decision-making. Figure 14 shows the (log) predicted standard deviations for different objectives on the approximate Pareto front of DTLZ2. According to the results in Figure 14(a)(b)(c), we can observe that the PSL model has low predicted standard deviations for almost all locations on the approximate Pareto front except the top corner and the top left boundary. From the log standard deviations in Figure 14(d)(e)(f), we find that PSL has a higher overall uncertainty level for objective 1/2 than objective 3, where the uncertainty distributions are quite similar for the first two objectives.

## F.9 Performance with different acquisition functions

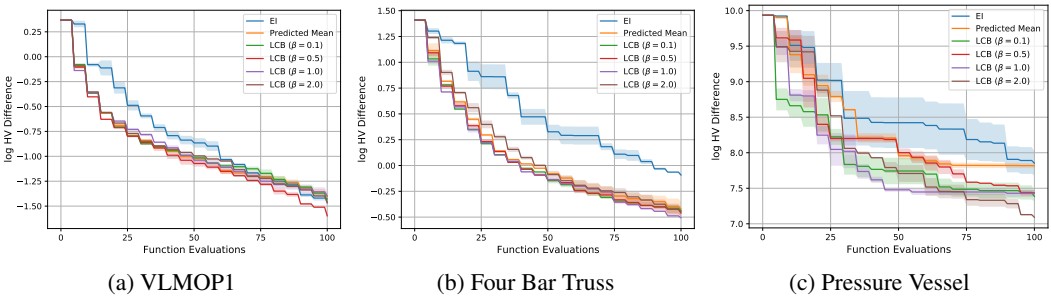

| (a) VLMOP1 | (b) Four Bar Truss | (c) Pressure Vessel |

Figure 15: The log hypervolume difference w.r.t. the number of expensive evaluation for PSL with different acquisition functions.

For the proposed PSL method, we can use different acquisition functions as the surrogate value for the Pareto set learning approach. Here, we compare the performance with 6 different acquisition functions/settings, namely, the Expected Improvement (EI), Predicted Mean, and the Lower Confidence Bound (LCB) with $\beta = 0.1, 0.5, 1.0$ and $2.0$ on four benchmark and real-world application problems.

According to the results shown in Figure 15, PSL with all 6 acquisition functions has reasonably good performance. Although PSL with EI performs slightly worse than the other acquisitions, it can still outperform many other MOBO algorithms with the results shown in Figure 5 of the main paper. These results confirm that learning the approximate Pareto set during optimization could be beneficial for MOBO.

Among the other acquisition functions and settings, no single choice can achieve the best performance for all problems. The predicted mean acquisition can be seen as LCB with $\beta = 0$, which is a pure exploitation strategy without considering the uncertainty (e.g., predicted variance) for exploration. It might be surprising that this greedy acquisition can still have a pretty good performance. Indeed, this greedy and purely exploitative approach has been recently studied and shown to have promising performance for single-objective BO [71, 18]. DGEMO [53] also uses the predicted mean as an acquisition function for MOBO. Garnett [30] briefly discusses this pure exploitative strategy at the end of Section 7.10 (posterior mean acquisition function and lookahead). One limitation of this greedy approach is that it might be stuck at a non-optimal location with no well-fitted surrogate model due to overexploitation [30]. There are many acquisition policies for Bayesian optimization, but finding the most suitable one for a given problem is still an open question [30], which is also the case for our PSL method.

## F.10  Results on problems with larger decision space

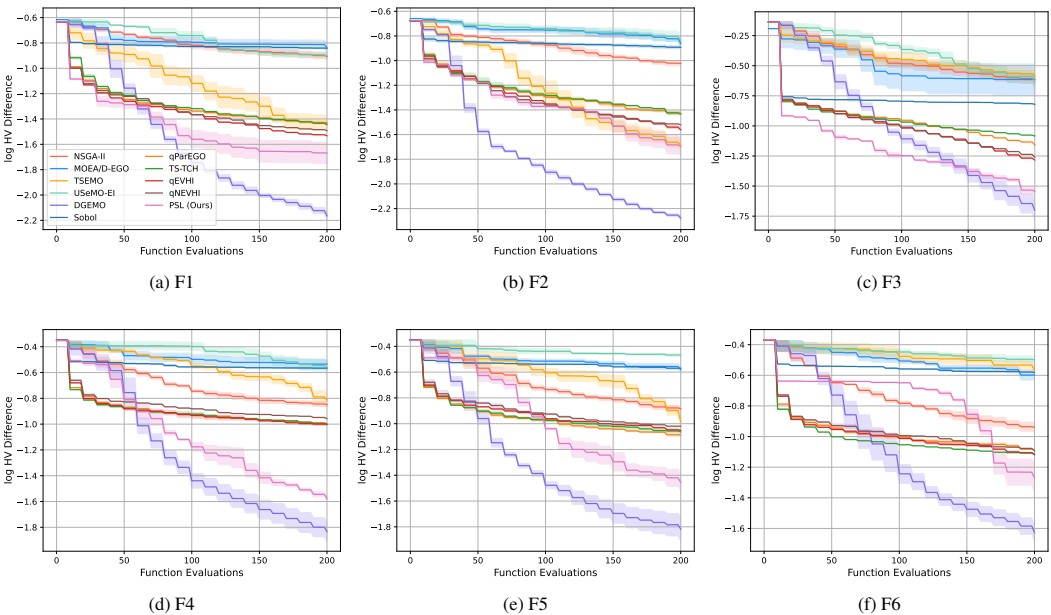

Figure 16: The log hypervolume difference w.r.t. the number of expensive evaluation for different algorithms for F1-F6 problems with 20 dimensional search space. The solid line is the mean value averaged over 10 independent runs for each algorithm, and the shaded region is the standard deviation around the mean value. **The label of each algorithm can be found in subfig(a).**

We conduct experiments on more challenging problems proposed in Appendix E.1 with a 20-dimensional decision space to further validate our proposed PSL method. The Pareto sets of these problems are all different complicated space curves in the search space. We set the batch size to 10, and run all algorithms with 20 batched evaluations. In other words, the evaluation budget is 200 for all problems. The experimental results in Figure 16 show that our proposed PSL method still has a promising performance on these problems.

According to the experimental results, the DGEMO [53] performs the best on these problems with larger decision space. The DGEMO algorithm has a well-designed local search strategy, which fully leverages the first and second order derivatives of the Gaussian process surrogate model to enrich its large candidate set for selection. In addition, it also explicitly encourages exploring diverse solutions during the optimization process. In contrast, for problems with larger decision space, our proposed PSL method could learn an inaccurate Pareto set, which could mislead and slow down the search process. In this work, we focus on the main idea of learning the Pareto set for MOBO problems, and do not aggressively include other powerful methods into our proposed algorithm. Its performance might be further improved by considering the second-order derivatives of GPs and diversity-guided batch selection as in DGEMO [53].

## F.11  Performance with Different Batch Sizes

We conduct additional experiments with different batch size ($B = 10$) as well as sequential evaluation ($B = 1$) for all problems. The total number of evaluations is still $10 + 100 = 110$ for all experiments. In other words, these settings have 10 and 100 batched evaluations, respectively. The experimental results are shown in Figure 17 and Figure 18. According to the results, our proposed PSL method has a better or comparable performance with other MOBO methods in most experiments. It also consistently outperforms the model-free scalarization methods with a large performance gap. These promising performances, along with the efficient run time, confirm that it is worth building the Pareto set model for expensive multi-objective optimization problems.

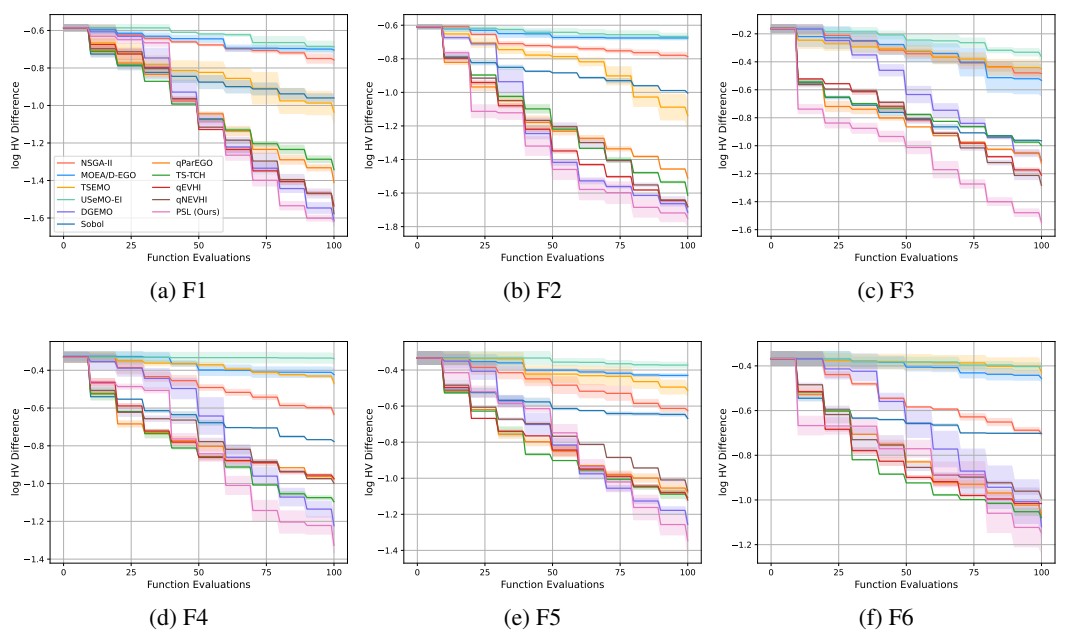

Figure 17: **Experimental results with Batch Size 10** on the log hypervolume difference w.r.t. the number of expensive evaluation for different algorithms. **The label of each algorithm can be found in subfig(a).**

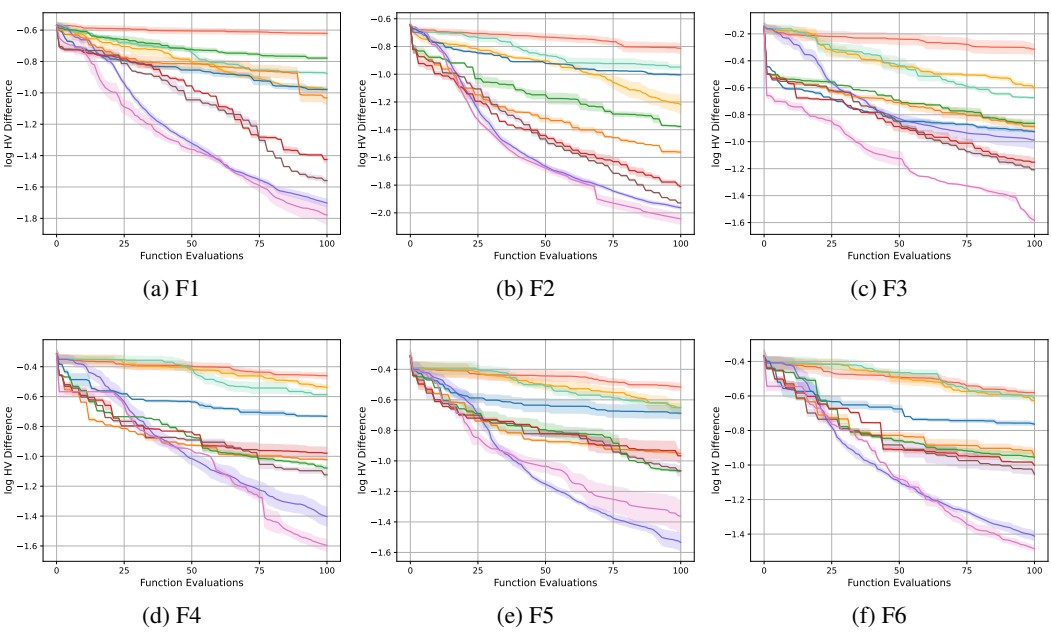

Figure 18: **Experimental results with Batch Size 1** on the log hypervolume difference w.r.t. the number of expensive evaluation for different algorithms.

# G Licenses

Table 4 lists the licenses for the codes and problem suite we used in this work.

Table 4: List of licenses for the codes and problem suite we used in this work.

| Resource | Type | Link | License |
|---|---|---|---|
| PSL-MOBO (Ours) | Code | https://github.com/Xi-L/PSL-MOBO | MIT License |
| Botorch[5] | Code | https://botorch.org/ | MIT License |
| DEGMO[53] | Code | https://github.com/yunshengtian/DGEMO | MIT License |
| pymoo[8] | Code | https://pymoo.org/ | Apache License 2.0 |
| reproblems[85] | Problem Suite | https://ryojitanabe.github.io/reproblems/ | None |