# OpenReview forum: "Pareto Set Learning for Expensive Multi-Objective Optimization"
_NeurIPS.cc/2022/Conference — NeurIPS 2022 Accept_

### Official Review · Reviewer_YcbD · 2022-06-20

**Rating:** 7
**Confidence:** 5
**Soundness:** 3 good
**Presentation:** 3 good
**Contribution:** 3 good

**Summary:**

In the context of expensive multi-objective black-box optimization, this article proposes to construct a surrogate model of the Pareto set. It is a neural network built upon surrogate models of the objectives, here Gaussian processes. The interest is first to help decision makers to navigate the estimated Pareto set and front. Then it is useful for selecting batches on candidates for sequential optimization, using the existing expected hypervolume improvement. The method is benchmarked over different classical test functions, against appropriate baselines.

**Questions:**

Could you show the learned Pareto front and set on a case where they are disconnected?
Appendix line 57: Is it still Expected improvement in this form?
Appendix C5: could you define the what ''uncertainty'' means here?

**Limitations:**

Limitations are discussed.

**Strengths And Weaknesses:**

Strength:
- the method integrates the structure of the Pareto set to help optimize the qEHVI criterion;
- the Pareto set model is useful for working with practitioners. The method to build it with backpropagation is original.
- the results are convincing in terms of speed and sample efficiency.

Weaknesses:
- models for the Pareto sets have been proposed in the multi-objective literature.
- it uses the qEHVI criterion.
- the behavior on disconnected Pareto front is not shown.
- a few aspects could be clarified, see Questions below.

---

> ### Author Response · Authors · 2022-08-02
> **Response to Reviewer YcbD**
>
> Thank you very much for your time and effort in reviewing our work. We are very glad to know you find our method can integrate the structure of the Pareto set to help MOBO, the proposed Pareto set model is useful for practitioners, the method is original, and the results are convincing in terms of speed and sample efficiency.
>
> We address your concerns as follows.
>
> > **1. Models for the Pareto Sets are Not New**
>
> Agree. Our main contribution is about how to learn an approximate Pareto set for expensive multi-objective black-box optimization problems. As the reviewer noticed, novel strengths of our proposed method are (1) it can help decision-makers to navigate the estimated Pareto set and front for better decision-making, and (2) it is good for better multi-objective Bayesian optimization in terms of speed and sample efficiency.
>
> > **2. Hypervolume Improvement for Batch Selection.**
>
> The reviewer pointed out that our proposed method also uses the hypervolume improvement criterion for batch selection. A key difference between our method and other hypervolume improvement approaches is that it only searches on an approximate Pareto set. We would like to add the following comments on EHVI.
>
> Efficiently finding a batch of solutions to optimize the EI/LCB of HVI could be challenging. qEHVI [1] and qNEHVI [2] are two promising approaches along this direction. From the point of view of optimizing HVI, our proposed method restricts the search procedure only on an approximate Pareto set, which is a low-dimensional (m-1) manifold in the decision space. Therefore, we can use a simple two-stage sample-then-select approach to find the batch of solutions for evaluations. Ideally, the set of solutions that optimize HVI should all be on the Pareto front. However, we agree with the reviewer that if an efficient method exists, directly optimizing the EI/LCB of HVI on the whole search space should be a principled approach for batch selection.
>
> On the other hand, the scalarization-based approach could have a close relationship with the hypervolume [3]. It is worthwhile studying how to leverage this relation for designing a more efficient algorithm, such as learning the whole Pareto set while inference the location of solutions (on the learned Pareto set) to optimize HVI at a single stage. It will be our future work.
>
> We have added this discussion on hypervolume improvement for batch selection in Appendix A.2.
>
> > **3. Disconnected Pareto Set**
>
> Thank you for your suggestion. We have shown the learned Pareto set for the ZDT3 problem with a disconnected Pareto set in Appendix F.6. For problems with a continuous search space (e.g., ZDT3), our proposed method can find an approximate continuous set that contains the disconnected parts of the Pareto set. How to design a more powerful Pareto set model to capture the disconnected patterns of the Pareto set (e.g., maybe with search region decomposition) is worthwhile studying in the future.
>
> > **4. Clarification**
>
> Thank you for pointing them out.
>
> ***Appendix line 57: Is it still Expected improvement in this form?***
>
> No, it is not. In this case, we first learn an approximate Pareto set with EI as the surrogate value for each scalarization, and then use the hypervolume improvement of LCB (with respect to those already-evaluated solutions) as the criteria to select a batch of solutions for evaluation.
>
> The main reasons for this choice are: 1) we want to avoid the (repeatedly) time-consuming Monte Carlo integration for calculating the expected hypervolume improvement; 2) the LCB (or the posterior mean only) is on the same scale as the value of those already-evaluated solutions, which make the calculation of HVI meaningful. We have rewritten this part to make this point clear.
>
> On the other hand, the scalarization-based approach could have a close relationship with the hypervolume [3]. In future work, we will study how to better leverage this relation for designing a more efficient algorithm, such as learning the whole Pareto set while inference the location of solutions (on the learned Pareto set) to optimize HVI at a single stage.
>
> ***Appendix C5: could you define what ''uncertainty'' means here?***
>
> Here the uncertainty means the standard deviation of the surrogate Gaussian process model for each objective on the approximate Pareto front. We have added the definition in this subsection accordingly.
>
> > **5. Reference**
>
> [1] S. Daulton, M. Balandat, and E. Bakshy. Differentiable expected hypervolume improvement for parallel multi-objective Bayesian optimization. NeurIPS 2020.
>
> [2] S. Daulton, M. Balandat, and E. Bakshy. Parallel Bayesian optimization of multiple noisy objectives with expected hypervolume improvement. NeurIPS 2021.
>
> [3] R. Zhang. and D. Golovin. Random hypervolume scalarizations for provable multi-objective black box optimization. ICML 2020.

---

> > ### Comment · Reviewer_YcbD · 2022-08-07
> > **Reply**
> >
> > Thank you for properly answering my comments and clarifying these points.

---

> > > ### Author Response · Authors · 2022-08-09
> > > **Thank You**
> > >
> > > Thank you for your reply and support. We will continually improve our work and properly reorganize the additional materials in the final version with one extra page.

---

### Official Review · Reviewer_HJkG · 2022-07-11

**Rating:** 5
**Confidence:** 4
**Soundness:** 2 fair
**Presentation:** 3 good
**Contribution:** 2 fair

**Summary:**

This paper presents a new Pareto set learning method to approximate the whole Pareto set for expensive multi-objective Bayesian optimization. it first generated a model that maps any trade-off preferences to their corresponding Pareto solutions with scalarization, then uses a batched acquisition search with the learned Pareto set model.

**Questions:**

The setting of the test problems is strange. The experiments on other test problems from the ZDT and DTLZ suites need to be added.

**Limitations:**

The impact of the aggregation function on the algorithm needs to be analyzed.

**Strengths And Weaknesses:**

The expensive multi-objective Bayesian optimization problems studied in this paper are a long-standing concern in the field of Bayesian optimization, and this paper presents a new algorithm for this. The main problem is that all the problems tested in the experiments have been proven to be quite simple. Moreover, the dimensionality of the test problem does not exceed 6. These raise concerns about the effectiveness of the algorithm proposed in this paper because the tested problems are too simple.

---

> ### Author Response · Authors · 2022-08-02
> **Response to Reviewer HJkG**
>
> Thank you very much for your time and effort in reviewing our work. We are glad to know you find our proposed method is a new algorithm for the expensive MOBO, which is a long-standing concern for Bayesian optimization.
>
> We address your concerns as follows.
>
> > **1. Main Concern: Tested Problems are Simple**
>
> Thank you for raising this concern.
>
> **Choice of Problems:** We mainly followed the most relevant MOBO works (e.g., qEHVI[1], qNEHVI[2], DGEMO[3]) to choose the problems for experimental comparison. The chosen combination of synthetic problem + real-world engineering design problem, and the problem settings (e.g., dimension, evaluation budget) are also consistent with these most relevant works.
>
> **More Experiments:** We have conducted new experiments on more challenging problems to further support our contribution in the revised paper. We propose 6 new test problems in Appendix E.2, of which the Pareto sets are all different space curves in the search space. We have done the experiments with two different settings: 1) 6-dimensional problems with 100 evaluations, and 2) 20-dimensional problems with 200 evaluations. The experimental results in Appendix F.1 confirm that our proposed PSL method can find good approximations to all Pareto sets and perform very well on these problems.
>
> **Scalability for MOBO:** The scalability, with the number of objectives or the dimensions of search space, is indeed a main challenge for the current MOBO algorithms. Our proposed method works quite well on the 20-D new test problems (Appendix F.1) and 20-D ZDT problems (Appendix F.7), but not on the 10-objective 50-D DLTZ2 problem.
>
> In future work, it could be interesting to investigate how to make PSL more scalable to those problems. One possible approach we can think of is to leverage the trust region method as in [4]. We provide more discussion in Appendix B on this limitation.
>
> > **2. Problem Setting and ZDT**
>
> We would like to point out that some problems in the ZDT/DTLZ suites are indeed not suitable for testing algorithms for expensive optimization. Therefore, most relevant MOBO works only choose some of the ZDT/DTLZ for experimental comparison for various reasons. For example, qEHVI[1] only uses DTLZ2, qEHVI[2] uses ZDT1 and DTLZ2, DGEMO[3] uses ZDT1-3 and the two-objective version of DTLZ.
>
> We have added the experiments on ZDT4/6 and DTLZ1 in Appendix F. The results show that the proposed PSL method can achieve promising results on the ZDT6 problem, while none of the MOBO algorithms can have satisfactory performance for ZDT4 and DTLZ1. One possible reason is that the ZDT4/DTLZ1 problems have a complicated optimization landscape (e.g., with a large number of local optimums). With a limited evaluation budget, the MOBO methods all fail to build a good surrogate model for such problems, and hence have poor performance.
>
> As pointed out by Reviewer MK65, the ZDT problems have a simple Pareto set, which is a line on the boundary of the decision space, and most decision variables of all Pareto solutions share the same values (e.g., 0). These properties could make ZDT an inappropriate benchmark for comparing different MOBO algorithms. In the next revision, we plan to move the results on the newly proposed 6 new problems into the main paper to replace the ZDT results.
>
> > **3. Impact of the Aggregation Function**
>
> We choose the weighted Chebyshev scalarization mainly due to its good theoretical property. Following your suggestion, we have now added an ablation study on the impact of the aggregation function v.s. the Pareto set model in Appendix F.5. The results confirm that the aggregation function itself is not special, and the proposed Pareto set model is important for the overall promising performance.
>
>
> > **4. Reference**
>
> [1] S. Daulton, M. Balandat, and E. Bakshy. Differentiable expected hypervolume improvement for parallel multi-objective Bayesian optimization. NeurIPS 2020.
>
> [2] S. Daulton, M. Balandat, and E. Bakshy. Parallel Bayesian optimization of multiple noisy objectives with expected hypervolume improvement. NeurIPS 2021.
>
> [3] M. K. Lukovic, Y. Tian, and W.Matusik. Diversity-guided multi-objective Bayesian optimization with batch evaluations. NeurIPS 2020.
>
> [4] S. Daulton, D. Eriksson, M. Balandat, and E. Bakshy. Multi-objective Bayesian optimization over high-dimensional search spaces. arXiv:2109.10964.

---

> ### Author Response · Authors · 2022-08-09
> **Further Response to Reviewer HJkG**
>
> Thank you again for your time and effort in reviewing our work. Following your and other reviewers' valuable comments and suggestions, we've carefully revised our paper to include new discussions (3 new sections in Appendix A/B/C) and many new experiments (8 new subsections in Appendix F) to address the raised concerns.
>
> There is only less than 12 hours left to the rebuttal deadline, and we sincerely want to know whether our responses can successfully address all your concerns. Please also let us know if you have further concerns. We are glad to continually improve our work to address them.

---

### Official Review · Reviewer_MK65 · 2022-07-11

**Rating:** 5
**Confidence:** 4
**Soundness:** 2 fair
**Presentation:** 3 good
**Contribution:** 2 fair

**Summary:**

This work proposes a novel method for multi-objective Bayesian optimization (MOBO). The work argues that the proposed method achieves greater sample efficiency and better runtime than existing alternatives. The proposed method leverages the structure of the Pareto set, acknowledging that the Pareto set is typically infinite and the often the result of optimization is a finite approximation.

**Questions:**

Why is using NN policy for candidate generation a good idea? What is the benefit?
  * The NN serves as a way to generate candidates. Why not just optimize the acquisition function under one scalarization (or more for generating a batch)?
  * Can you provide an ablation study that removes the NN and simply optimizes the acquisition function? The importance of each the two main contributions---1) using augmented Tchebyshev scalarization with an ideal point and epsilon and 2) using a NN to predict the optimal designs---is currently not clear.
* Regarding batch selection: qParEGO also selects different scalarizations, randomly, for each point in the batch. The proposed batch selection criteria of using HVI (based on the predicted LCB) to select points is not well-motivated. Why not just maximize expected improvement or LCB of HVI? The choice of using HVI for batch selection is a large departure from proposed approach for sequential optimization and is not well-justified
* This work is missing an introduction to Bayesian optimization, which would make it hard for a non-BO expert to read this paper.
* Hypervolume improvement is introduced before hypervolume. I recommend switching this ordering (and potentially introducing these concepts earlier)
* “It would be computationally intensive to find a set to exactly maximize the hypervolume improvement (12). “ This is commonly done in expected hypervolume improvement, albeit with a sequential greedy approximation. I recommend clarifying this statement to state that jointly maximizing B candidates is difficult and therefore sequential greedy selection is typically used (Daulton et al., 2020).
* The pareto set of designs for the ZDT problems exist in a tiny slice of the search space. This biases performance towards the PSL based approach where a NN could very well predict the optimal designs in they are close to together and at least one has been observed. Furthermore, this makes many improvement-based acquisition function hard to optimize. Therefore, using the proposed epsilon adjustment has outsized impact, but this could be done for any acquisition function. In addition, the initial designs used for acquisition optimization are particularly important as often these will be zero. Hence, acquisition optimization is sensitive to the initialization and techniques such as perturbing the best designs observed so far with a small amount of zero-mean Gaussian noise significant influence optimization performance. This feature if ZDT is particularly important because one contribution of this work is the method for generating candidates using PLS with circumvents the classical acquisition optimization. Heuristics such as perturbing the best points with a small amount of zero-mean Gaussian noise should be compared against to understand this paper's contribution.
* Appendix, L198: “qEHVI needs an extremely long runtime  to calculate the expected hypervolume improvement for more than 3 objectives at each step.” Daulton et al., 2020 run qEHVI on vehicle safety with 3 objectives and Daulton et al., 2021 run qEHVI and qNEHVI on car side impact with 4 objectives.
* Why is LaMoo (Zhao et al., 2022) not compared against?
* Why is qNEHVI (Daulton et al., 2021) not compared against, since is scales better with the batch size than qEHVI?
* Line 76: “expectation improvement” -> expected improvement
* Line 126: Figure 1b does not show “our proposed PSL method also learns an estimated Pareto set Mpsl with the predicted Pareto front fˆ (Mpsl ) to approximate the Pareto set Mps and Pareto front f (Mps ) “
* Line 250: “\hat{f} are the surrogate values” – it is unclear whether this is the posterior mean or the LCB?



**Limitations:**

The paper briefly touches on limitations in the last paragraph, but those limitations are for MOBO in general rather than focusing on this methodology. The paper would benefit from a discussion of worst-case behavior or situations were the method might fail or struggle would be good to include. Negative societal impacts are not discussed.

**Strengths And Weaknesses:**

The problem of multi-objective optimization of expensive function is important. However, many existing works have already tackled this problem, as noted in the paper. Hence, subsequent efforts can at best achieve incremental gains. The method amounts to 1) using the known connection between an augmented TCH scalarization and pareto optimal points, 2) training a neural network select the next candidate designs (these would be believed to be Pareto optimal if the \hat{f} were set to the posterior mean, but instead this work sets it to a lower confidence bound).

The paper is generally well-written, but the augmented Chebyshev scalarization is already used by ParEGO with EI. It is unclear why a NN should be used here to select designs (the main novel component). The empirical results show that the proposed methods works quite well, but raises questions. Given that there are no theoretical results, the burden is on the empirical validation and the motivation for the methodology, but both currently fall short of making a compelling case for the utility of this method.

---

> ### Author Response · Authors · 2022-08-02
> **Response to Reviewer MK65 [3/3]**
>
> > **10. Comparison with qNEHVI**
>
> Following your suggestion, we have compared with qNEHVI on the 6 new test problems as shown in Appendix F.1. The results show that qNEHVI performs similarly to qEHVI on all problems. We will add the experimental results of qNEHVI in all problems (along with the runtime analysis) in the next revision.
>
> > **11. LaMOO**
>
> We do not directly compare with LaMOO since it is a meta-algorithm, which can be used with other multi-objective optimization algorithms, such as LaMOO + NSGA-II and LaMOO + qEHVI. Similarly, the trust region method [6] can be combined with sequential greedy HVI to tackle the challenging multi-objective optimization problems with high-dimensional search space [7].
>
> Studying how to efficiently combine LaMOO/trust-region approach with the proposed PSL method will be our future work. We have now added a short discussion in Appendix B.3.
>
> > **12. More Discussion on the Limitation**
>
> Following your suggestion, we have added a new limitation and potential future work section in Appendix B to discuss: 1) the approximate Pareto set could be inaccurate,  2) defining the user preference is challenging for black-box optimization, and 3) scalability with the high-dimensional search space.
>
> > **13. Potential Societal Impact**
>
>
> Thank you for pointing this out. We have added a new subsection in Appendix C to discuss the potential societal impact of our proposed method. A major positive impact could be the flexible decision-making with an approximate Pareto set. On the negative side, the leakage of the Pareto set model might unintentionally reveal the problem information and user preference, which should be avoided. We have also mentioned the risk that an inaccurate approximate Pareto set will bring to the application.
>
> > **14. Typos and Unclear Description**
>
> Thank you for pointing them out. We have corrected the typos and modified the unclear description, and will carefully proofread the whole manuscript.
>
> **For hat f**: We can use the posterior mean or other acquisition (e.g., LCB, EI) as the surrogate values in the proposed PSL method. An ablation study of the choice of different surrogate values has been provided in Appendix F.3. In the main paper, we use the LCB as the surrogate value. We have added a short description to make this point clear.
>
> > **15. Reference**
>
>
> [1] S. Daulton, M. Balandat, and E. Bakshy. Differentiable expected hypervolume improvement for parallel multi-objective Bayesian optimization. NeurIPS 2020.
>
> [2] M. K. Lukovic, Y. Tian, and W.Matusik. Diversity-guided multi-objective Bayesian optimization with batch evaluations. NeurIPS 2020.
>
> [3] S. Daulton, M. Balandat, and E. Bakshy. Parallel Bayesian optimization of multiple noisy objectives with expected hypervolume improvement. NeurIPS 2021.
>
> [4] R. Zhang. and D. Golovin. Random hypervolume scalarizations for provable multi-objective black box optimization. ICML 2020.
>
> [5] R. Garnett. Bayesian Optimization. Cambridge University Press, 2022. In Preparation.
>
> [6] D. Eriksson, M. Pearce, J. R. Gardner, R. Turner, and M. Poloczek.
> Scalable global optimization via local Bayesian optimization. NeurIPS 2019.
>
> [7] S. Daulton, D. Eriksson, M. Balandat, and E. Bakshy. Multi-objective Bayesian optimization over high-dimensional search spaces. arXiv:2109.10964.

---

> > ### Comment · Reviewer_MK65 · 2022-08-06
> > **discussion**
> >
> > Thanks for the thorough response.
> >
> > > Therefore, the selected weighted scalarization(s) might be close to those already-evaluated solutions.
> >
> > This is a good point, but it is also not an issue for HVI based methods.
> >
> > > 2. Ablation Study with/without NN
> >
> > Thanks for adding this. However, PLS is still using HVI for batch selection, so I guess the relevant comparison would be maximizing HVI w/ and w/o the PLS model, which are already included in the main experiments. The lingering difference though that PLS uses the GP surrogate model predicted LCB for computing HVI. The ablation that would be of interest is maximizing HVI (using LCB) across the candidate set from the PLS model vs  across the entire design space (using gradients). In addition to log regret, it would be interested to see which approach yields higher HVI (better acquisition values). Particularly, on test problems like ZDT1, the initialization for selecting starting points for gradient-based optimization seems like it would be very important. What heuristic is used? Heuristics such as perturbing the best points with a small amount of zero-mean Gaussian noise should be compared against to understand the importance of the NN for generating a candidate set.
> >
> > >4. Other Benefits of the Pareto Set Model
> >
> > One could simply use NSGA-II to learn the pareto frontier by optimizing the posterior means  (or LCB) of the GP. I also don’t understand why a decision maker would want to understand the optimal solutions under LCB (after optimization). It seems much more logical to understand the optimal solutions under the posterior mean (after optimization).
> >
> > Lastly, what distributions are used over the preferences? This seems important for training the NN because the sampled preferences will dictate the composition of the training data for the NN. Particularly, if the sampled preferences are not uniform across the pareto frontier, then the training data may be quite skewed w.r.t to its coverage of the Pareto frontier (e.g. many of the designs x used for training may correspond to a small region of the pareto frontier in objective space)

---

> > > ### Author Response · Authors · 2022-08-09
> > > **Response to Discussion [3/3]**
> > >
> > > > **7. What distributions are used over the preferences? This seems important for training the NN because the sampled preferences will dictate the composition of the training data for the NN.**
> > >
> > > We agree with the reviewer that the preference distribution could be important for training the Pareto set model. In this paper, without any prior information, we simply use a uniform distribution over the whole valid preferences simplex (e.g., $\Lambda = \{\lambda \in R^m_{+}|\sum \lambda_i = 1\}$) for all problems. As discussed in Appendix B.1 and B.2, if the decision-maker's preference distribution is available, we can simply use it to learn the Pareto set and guide the optimization in PSL.
> > >
> > > > **8. Particularly, if the sampled preferences are not uniform across the Pareto frontier, then the training data may be quite skewed w.r.t to its coverage of the Pareto frontier (e.g. many of the designs x used for training may correspond to a small region of the Pareto frontier in objective space)**
> > >
> > > Agree, this is an important concern for the scalarization-based method. In this paper, we simply use the uniform distribution on the preference, but uniform distribution on the preferences does not necessarily mean the distribution on the Pareto front.
> > >
> > > To tackle this issue, one possible approach we can think of is to leverage the relationship between scalarization and hypervolume [4] to adaptively adjust the preference distribution. The goal is to let the corresponding approximate Pareto solutions can maximize the hypervolume, which means they should be uniformly distributed on the approximate Pareto front, so as for the training data.
> > >
> > >
> > > One step further could be to adaptively adjust the preference distribution to let corresponding solutions maximize the HVI with respect to those already-evaluated solutions. From the optimization perspective, this approach could combine our two-stage approach into a single adaptive stage. We will explore these interesting directions in future work.
> > >
> > > > **Reference**
> > >
> > > [1] Y. Zhao, L. Wang, K. Yang, T. Zhang, T. Guo, and Y. Tian. Multiobjective optimization by learning space partitions. ICLR2022.
> > >
> > > [2] S. Daulton, D. Eriksson, M. Balandat, and E. Bakshy. Multi-objective Bayesian optimization over high-dimensional search spaces. UAI2022.
> > >
> > > [3] M. K. Lukovic, Y. Tian, and W.Matusik. Diversity-guided multi-objective Bayesian optimization with batch evaluations. NeurIPS 2020.
> > >
> > > [4] R. Zhang. and D. Golovin. Random hypervolume scalarizations for provable multi-objective black box optimization. ICML 2020.

---

> > > > ### Comment · Reviewer_MK65 · 2022-08-09
> > > > **response 2**
> > > >
> > > > Thanks for the response. Most of my concerns have been addressed and I will increase my score.
> > > >
> > > > Regarding NSGA-II only recovering finite approximate Pareto sets. This is true, but an infinite PF does not seem to provide utility. Furthermore, with more objectives, more optimal designs are not necessarily a good thing. With more data points,  more comparisons must be made. These comparisons are difficult with many objectives and augmented Chebyshev scalarizations are a hard to reason about in many-objective spaces, so the PS model does not seem very helpful.

---

> > > > > ### Author Response · Authors · 2022-08-10
> > > > > **Thank You**
> > > > >
> > > > > Thank you for your further response and the increased score. We will further elaborate the discussion on the practicality of the approximate Pareto set in the final version (revision is now not allowed).
> > > > >
> > > > > We agree with the reviewer that making decisions/comparisons could be difficult with many conflicting objectives. In addition to the whole approximate PS, once the decision-maker choose initial acceptable solutions, the PS model can adaptively generate a local Pareto set that only contains similar solutions around them, and help the decision makers to further refine their choices. It could be easier to make decisions among solutions with similar and comparable trade-offs than to compare solutions with many very different objective values. The PS model can serve as a flexible interactive tool to support difficult decision making with many objectives.
> > > > >
> > > > > The PS model has a fast runtime similar to traditional multiobjective optimization algorithms, but it can additionally allow the decision-makers to navigate the whole approximate Pareto set. We believe this property and the other usages of the learned Pareto set can be further explored in future work.

---

> > > ### Author Response · Authors · 2022-08-09
> > > **Response to Discussion [2/3]**
> > >
> > > > **3. On test problems like ZDT1, the initialization for selecting starting points for gradient-based optimization seems like it would be very important. What heuristic is used?**
> > >
> > > Agree. Due to the simple Pareto set, the optimization performance on ZDT1 could be significantly affected by the initialization. In the experiment, we did not leverage this property for an unfair comparison on the ZDT problems. PSL randomly initializes the parameters of the Pareto set model at each step, and then uses the gradient-based method to learn the approximate Pareto set. In our approach, the initially learned Pareto set is a random curve in the search space far away from the boundary. Then it gradually converges to the Pareto set with the efficient gradient-based update.
> > >
> > > We have added an illustration in Appendix F.13, which shows how the approximate Pareto set is learned by PSL.
> > >
> > > > **4. Heuristics such as perturbing the best points with a small amount of zero-mean Gaussian noise should be compared against to understand the importance of the NN for generating a candidate set.**
> > >
> > > Following your suggestion, we have added an ablation on small perturbing the best points v.s. Pareto set model for generating the candidate set in Appendix F.14. Based on the results, the Gaussian perturbing method can improve the performance of simple scalarization on problems with simple Pareto set (e.g., ZDT1) but not the problems with complicated Pareto set (e.g., F4-F6). On all problems, our proposed PSL method still achieves the best performance.
> > >
> > > DGEMO [3] also has a local search approach to expand the candidate set around the best points (in term of surrogate value) with the first and second derivatives of the GP surrogate model. In our experiments, PSL can outperform DGEMO on most problems, which confirms the importance of the NN for generating the candidate set.
> > >
> > > On the other hand, the local search methods might provide complementary candidate solutions to PSL, especially at the early stage of optimization when the approximate Pareto set is inaccurate. We will investigate how to efficiently combine PSL with the local search approaches in future work.
> > >
> > > > **5. I also don’t understand why a decision maker would want to understand the optimal solutions under LCB (after optimization). It seems much more logical to understand the optimal solutions under the posterior mean (after optimization).**
> > >
> > > In PSL, the LCB acquisition is mainly for optimization, and we always provide the approximate Pareto set under the posterior mean to decision makers. In the paper, the learned Pareto sets in Figure 1/3/6/7 are all under the posterior mean. We have added short explanations in the paper to make this point clear (line 223-224, line 260, line 301-302).
> > >
> > > Since the runtime for building Pareto set model is very fast, it is also feasible to additionally provide the approximate Pareto set under the posterior mean (and with the standard deviation as in Appendix F.8) to the decision makers during the optimization process. This information could be useful for interactive optimization.
> > >
> > > > **6. Other Benefits: One could simply use NSGA-II to learn the pareto frontier by optimizing the posterior means (or LCB) of the GP.**
> > >
> > > The key difference is NSGA-II (and other multiobjective optimization algorithms) can only find a finite set of solutions to approximate the Pareto set which could have infinite solutions. In contrast, our Pareto set model can approximate the whole Pareto set, such that the decision-maker can easily explore any trade-off solutions on the Pareto front by adjusting the preference.
> > >
> > > The required number of finite solutions for a good approximation could increase exponentially with the number of objectives. For example, if $50$ solutions are required to well approximate the Pareto set/front for a two-objective problem (1-D curve), the number could be $50^2$ for a three-objective problem (2-D surface), and be general $50^{m-1}$ for an $m$-objective problem. It will be impractical for $m \geq 4$.
> > >
> > > The finite set of solutions may not contain the most preferred solutions for the decision-maker, while PSL allows the decision-maker to easily explore the whole approximate Pareto front to locate the most suitable solution(s). This property of PSL is also important to support the benefits of the approximate Pareto set as we discussed in the initial response.
> > >
> > > Given the fast running time and its own advantages, we believe the proposed Pareto set model could be a promising alternative to the current multiobjective optimization algorithm such as NSGA-II.

---

> > > ### Author Response · Authors · 2022-08-09
> > > **Response to Discussion [1/3]**
> > >
> > > Thank you very much for your response and the follow-up questions. We address them point by point as follows.
> > >
> > > To keep the order of the original subsections unchanged, we temporarily put all new experimental results and analyses in Appendix F.11 - F.14 (p31-33). All the new materials are highlighted in **blue**, and will be put properly in the final version.
> > >
> > > > **1. The ablation that would be of interest is maximizing HVI (using LCB) across the candidate set from the PLS model vs across the entire design space (using gradients).**
> > >
> > > **Ablation Study**
> > >
> > > Thank you for your suggestion. We have conducted an ablation study on maximizing the hypervolume improvement with LCB on the approximate Pareto set vs. searching across the whole design space. The experimental results on three newly proposed problems (i.e., F4, F5, and F6) with complicated Pareto sets can be found in Appendix F.11. We also provide the qEHVI results as reference.
> > >
> > > Based on these results, it is clear that our proposed PSL method can outperform directly searching the entire search space to maximize HVI with LCB. The qHVI-LCB method performs similarly to qEHVI, which indicates that different acquisitions (for HVI) do not have a significant impact on the optimization performance. In other words, with the learned Pareto set, our proposed PSL method can efficiently conduct the acquisition optimization mainly on the promising low-dimensional (e.g., (m-1)) manifold, and then lead to significantly better Bayesian optimization performance.
> > >
> > > **Reason for Promising Performance**
> > >
> > > One possible reason for this performance gap could be the difficulty of directly searching the whole design search for maximizing HVI. As shown in recent works [1,2], the performance of qEHVI can be improved using meta search region management algorithms such as LaMOO [1] and trust region methods [2]. It implies that directly searching the whole design space is not efficient.
> > >
> > > These meta-algorithms adaptively decompose the whole search region into different subregions along with the optimization process, while our PSL method directly learns the whole approximate Pareto set at each step. We will add the comparison with those meta-algorithms in the final version (LaMOO [1] is not open-sourced yet, and the code for the trust-region method [2] was just released several days ago). Studying how to efficiently combine PSL with these algorithms could be an important future work as discussed in Appendix B.
> > >
> > > Together with the discussion in our initial response, the advantages of PSL can be summarized from the following two viewpoints:
> > > - From the viewpoint of scalarization based methods, PSL proposes a novel two-stage approach to select a small batch of appropriate solutions from the approximate Pareto front, which takes those already-evaluated solutions into consideration. As a result, it significantly outperforms other scalarization-based methods.
> > > - From the viewpoint of hypervolume improvement based methods, in an ideal case, the set of solutions that optimize HVI should all be on the Pareto front. Restricting the search on the approximate Pareto set, PSL leads to an efficient HVI algorithm.
> > >
> > > > **2. In addition to log regret, it would be interested to see which approach yields higher HVI (better acquisition values).**
> > >
> > > Following your suggestion, we have added the experiment to compare the HVI-LCB acquisition values obtained by different search methods in Appendix F.12. We run our proposed PSL as the optimization algorithm, and conduct different search methods at each iteration. The compared search methods are: 1) searching across the entire design space, 2) our PLS search on the approximate Pareto front, and 3) the HVI-LCB acquisition values for the whole approximate Pareto front with 1,000 sampled solutions (which can be treated as the upper bound for our method).
> > >
> > > Based on the results, it is clear that our proposed PSL search can produce better HVI-LCB acquisition values than directly searching the entire design space.

---

> ### Author Response · Authors · 2022-08-02
> **Response to Reviewer MK65 [2/3]**
>
> > **4. Other Benefits of the Pareto Set Model**
>
> In addition to the improved sample efficiency for MOBO, our proposed Pareto set model can also provide valuable information to support decision-making.
>
> Please refer to our Response 1 to Reviewer KcfL (or Appendix A.3) for more details.
>
> > **5. Introduction to Bayesian Optimization**
>
> Thank you for your suggestion. We have added a brief introduction to Bayesian optimization and the Gaussian process in Section 3, and refer readers to a Bayesian optimization textbook [5] for a detailed introduction due to the page limit.
>
> > **6. The Order of Hypervolume and Hypervolume Improvement**
>
> Following your suggestion, we have switched the ordering, which now introduced the hypervolume first and then the HVI. Considering the page limit (no extra page is allowed during the response period), we keep these concepts in the current position (mainly for solution selection). We could further improve the exposition if you have any suggestions.
>
> > **7. Clarification on Expected Hypervolume Improvement and the Sequential Greedy Approximation**
>
> Following your suggestion, we have improved the clarification on this issue. It should be clear that jointly maximizing a batch of solutions to maximize EHVI is difficult, and the sequential greedy selection method is typically used for efficient optimization as in qEHVI [1].
>
> > **8. ZDT and Biased Performance**
>
> We fully agree with the reviewer that the ZDT problem has a simple Pareto set, which is a line on the boundary of the decision space, and most decision variables of all Pareto optimal solutions share the same values (e.g., 0). These properties could make ZDT problems not appropriate for comparing different MOBO algorithms. We chose the ZDT problem for comparison solely due to it is widely used in multi-objective optimization, and our proposed PSL method is not specially designed for such problems.
>
> Instead of conducting further analysis on the inappropriate ZDT problems, we choose to run a new comparison on more difficult test problems. We propose 6 new test problems in Appendix E.2, of which the Pareto sets are curves in the search space. The experimental results shown in Appendix F.1 confirm that our proposed PSL method works very well on these problems. We will move these results into the main paper (to replace ZDT problems) in the next revision.
>
> Together with the results on the other 4 synthetic and 5 real-world application problems in the main paper, the experiments demonstrate our contribution on sample efficiency for MOBO.
>
> > **9. qEHVI and qNEHVI on Problems with 3-4 Objectives**
>
> Thank you for pointing this out. We have also compared with qEHVI on different 3-objective problems (e.g., VLMOP3, DTLZ2, Disk Brake, Gear Train, and Rocket Injector) in the original submission. The mentioned example and illustration were to show that problems with many objective functions (e.g., 10) would be very challenging for expensive optimization, which is a general limitation for all MOBO algorithms.
>
> We will modify the current illustration to mention the original EHVI is time-consuming, and the improvement proposed in qEHVI and qNEHVI make it possible to properly handle problems with 3-4 objectives.

---

> ### Author Response · Authors · 2022-08-02
> **Response to Reviewer MK65 [1/3]**
>
> Thank you very much for your thorough and constructive comments and suggestions. We are glad to know you find our work novel and well-written.
>
> Our responses are as follows.
>
> > **1. Motivation:  NN Policy v.s. Simple Scalarization**
>
> As the reviewer noticed, our proposed NN-based Pareto set learning method has a close relation to the scalarization-based methods. But it can overcome a major disadvantage of the current scalarization methods for expensive optimization.
>
> The scalarization methods randomly select one (e.g., ParEGO) or a batch of scalarized subproblems (e.g., MOEA/D-EGO, TS-TCH, and qParEGO) at each iteration. By optimizing the acquisition function (e.g., EI, LCB, Thompson Sampling etc) on each scalarization, they generate a batch of solutions for expensive evaluation.
>
> A major limitation of all these scalarization methods is that they do not explicitly consider those already-evaluated solutions, neither for choosing the weighted scalarization, nor for selecting the next (batch of) solution(s) for expensive evaluation. Therefore, the selected weighted scalarization(s) might be close to those already-evaluated solutions. In other words, even the obtained solution(s), can maximize the EI/LCB for the selected scalarization, could still be similar to those already-evaluated ones, and indeed not an optimal choice for multi-objective optimization. This limitation leads to inferior performance of these scalarization methods, as reported in [1,2] and in our experimental results.
>
> To overcome this limitation, our proposed method has a two-stage approach:
>
> - Stage 1: Using an NN-based Pareto set model, it first efficiently samples a dense set of candidate solutions to cover the whole approximate Pareto front (of posterior mean, EI, or LCB etc).
> - Stage 2: Then it selects a small batch of appropriate solutions from this dense set for expensive evaluation. For selection, we use the HVI criteria to take both the selected and already-evaluated solutions into consideration.
>
> In this way, our proposed method can efficiently explore the whole approximate Pareto front, and choose the most appropriate solutions (on the approximate Pareto front, while far from the selected and already-evaluated ones) for expensive evaluation. The experimental results have validated the efficiency of the proposed method.
>
> We have added a discussion on this motivation in Appendix A.1.
>
> > **2. Ablation Study with/without NN**
>
> As noticed by the reviewer, without the NN model, the algorithm is the augmented Chebyshev scalarization with an ideal point and epsilon, which is similar to ParEGO (for one solution) or MOEA/D-EGO/TS-TCH (for a batch of solutions). We choose this scalarization mainly due to its good theoretical property.
>
> Following the reviewer's suggestion, we have added an ablation study on the proposed method with/without NN in Appendix F.5. The results confirm that the proposed NN-based Pareto set model is important for the overall promising performance.
>
> > **3. Why HVI for Batch Selection**
>
> As discussed above, the current scalarization approach (e.g., (q)ParEGO) randomly selects different scalarization(s) at each iteration, which is to randomly generate a small set of sparse solutions on the approximate Pareto set (with EI, LCB) for evaluation. In other words, the selected solutions are a small random subset of the whole approximate Pareto set (our Stage 1).
> In our proposed algorithm, we use the HVI to select a small batch of solutions from the dense approximate Pareto set, which can be aware of both selected and already-evaluated solutions and leads to better optimization performance (our Stage 2).
>
> Efficiently finding a batch of solutions to optimize the EI/LCB of HVI could be very challenging. qEHVI [1] and qNEHVI [3] are two promising approaches along this direction. From the viewpoint of optimizing HVI, our proposed method restricts the search procedure only on the approximate Pareto set, which is a low-dimensional (m-1) manifold in the decision space. Therefore, we can use a simple two-stage sample-then-select approach to find the batch of solutions for evaluations. For the optimal situation, the set of solutions that optimize HVI should all be on the Pareto front. However, we agree with the reviewer that if an efficient method exists, directly optimizing the EI/LCB of HVI on the whole search space should be a principled approach for batch selection.
>
> On the other hand, the scalarization-based approach could have a close relationship with the hypervolume [4]. It will be very interesting to study how to better leverage this relation for designing a more efficient algorithm, such as learning the whole Pareto set while inference the location of solutions (on the learned Pareto set) to optimize HVI at a single stage. These will be our future work.
>
> We have added this discussion in Appendix A.2.

---

### Official Review · Reviewer_KcfL · 2022-07-13

**Rating:** 8
**Confidence:** 5
**Soundness:** 3 good
**Presentation:** 4 excellent
**Contribution:** 3 good

**Summary:**

The paper proposes an algorithm for batch multi-objective Bayesian optimization. The acquisition function uses Chebyshev scalarization. The goal is to solve problems with an infinite size pareto optimal set and find a pareto manifold of solutions.

**Questions:**

Several applications have an infinite pareto set of solutions. However, in practice, having access to an approximation of the manifold over the input space in the context of expensive evaluations, might not be necessarily useful. Given the cost of the objectives, the budget might be exhausted by the time a good approximation of the manifold is built. At that point of the optimization process, access to the pareto set manifold would not be useful without access to output values. Also relying on only the approximate input space manifold can be very risky in several applications given the uncertainty of the prediction.

**Limitations:**

Yes the authors discussed limitations briefly. I suggest including a discussion about the challenges in defining the preference over objectives in the context of black-box functions

**Strengths And Weaknesses:**

Strengths

+The problem setting is interesting. Continuous pareto set There are several real-world problems with a manifold of solutions. The Idea of creating a projection of the manifold based on the preference defined by the practitioner is appealing.
+ the approach is well structured and novel
+ The paper is well written, and the proposed solution is clear.
+ The paper covered lots of baselines on several synthetic and real-world problems

Weaknesses:

+ The practicality of accessing the pareto set manifold in the context of expensive experiments might be questionable.
+ Defining user preference in terms of scalars is challenging in practice and there is no clear way on how to do it
+ The experiment comparison is missing an important relevant algorithm from the preference over objectives literature [1]
+ The contribution of the batch selection is minor and incremental

[1] Abdolshah, Majid, et al. "Multi-objective Bayesian optimization with preferences over objectives." Neurips (2019).

---

> ### Author Response · Authors · 2022-08-02
> **Response to Reviewer KcfL [2/2]**
>
> > **2. Defining User Preference is Challenging for Black-box Optimization**
>
> Totally agree. Although some scalarization methods (e.g., Chebyshev scalarization) have good theoretical properties to connect the scalars to their corresponding Pareto solutions, it could be not so easy to define user preferences in terms of scalars, especially in the black-box expensive optimization setting. Some efforts have been made to tackle the preference assignment problem [3,4,5].
>
> In many applications, the decision-makers might even do not know their actual preferences before making their decision. If the approximate Pareto set can be properly learned, instead of asking users to provide their preferences in terms of fixed scalars, our proposed approach can let them interactively explore the whole approximate Pareto set/Pareto front to select their most preferred solutions. In this interactive way, it could be much easier for the decision-makers to accurately express and assign their preferences.
>
> However, it could be difficult to precisely obtain user preference if a good approximate Pareto set is not available (e.g., in the early stage of optimization, or without enough budget). We have added more discussion on this challenge in Appendix B.2.
>
> > **3. Missing Experiment Comparison with Preference-based Algorithm [3]**
>
> Thank you for raising this issue. The MOBO-PC algorithm proposed in [3] is an elegant and promising way to incorporate preference into the MOBO process via the preference-order constraints, which does not require any prior knowledge on the (approximate) Pareto front. In our paper, we mainly validate our proposed PSL algorithm for learning the whole Pareto set/front, so the results of the preference-based approach (e.g., [3]) are not included.
>
> Following your suggestion, we have tried to add a direct comparison with MOBO-PC on preference-based optimization, but failed to reproduce the results for MOBO-PC. We have sent an email to the author of [3], and will add comparison results once the code for MOBO-PC is available.
>
> The preference-order constraint proposed in [3] has a close relationship with the lexicographic approach for multi-objective optimization [6], which is a non-scalarizing method with good theoretical property (e.g., connection to weakly Pareto optimal solution). How to incorporate this information into our proposed Pareto set learning model for more flexible preference incorporation could be interesting for future work.
>
> > **4. Contribution of Batch Selection is Incremental**
>
> The main novelty/contribution of the proposed method is to learn the approximate Pareto set/front to (1) support flexible decision making, and (2) improve the sample efficiency for MOBO. The proposed batch selection method is a simple way to select solutions from the learned Pareto set.
>
> We agree with the reviewer that the contribution of the batch selection alone could be minor and incremental. But the Pareto set learning approach behind the batch selection could be crucial to achieving promising optimization performance. Please also see our response to Reviewer MK65 (Response 123) for a detailed explanation.
>
> > **5. Limitation: Preference Definition for Black-Box Optimization**
>
> Thank you for your suggestion. We have added a discussion on this challenge in Appendix B.2 for limitations and potential future work.
>
> > **6. Reference**
>
> [1] G. Malkomes, B. Cheng, E. H. Lee, and M. Mccourt. Beyond the pareto efficient frontier: Constraint active search for multiobjective experimental design. ICML 2021.
>
> [2] R. Garnett. Bayesian Optimization. Cambridge University Press, 2022. in preparation. (Page 294, Section 12.6 What's Next? Bullet Point 3)
>
> [3] M. Abdolshah, A. Shilton, S. Rana, S. Gupta, and S. Venkatesh. Multi-objective Bayesian optimisation with preferences over objectives. NeurIPS 2019.
>
> [4] R. Astudillo and P. Frazier. Multi-attribute Bayesian optimization with interactive preference learning. AISTATS 2020.
>
> [5] B. Paria, K. Kandasamy, and B. Póczos. A flexible framework for multi-objective Bayesian optimization using random scalarizations. UAI 2020.
>
> [6] C. Romero. Extended lexicographic goal programming: a unifying approach. Omega: The International Journal of Management Science 2001.

---

> > ### Comment · Reviewer_KcfL · 2022-08-07
> > **Response**
> >
> > Thank you for the detailed answer and for updating the manuscript accordingly.
> >
> > I believe that approach is novel in its context and advances the state of the art for scalarization-based bayesian optimization. I increased my score.

---

> > > ### Author Response · Authors · 2022-08-09
> > > **Thank You**
> > >
> > > Thank you for your response and support. We will continually improve our work and properly reorganize the additional materials in the final version with one extra page.

---

> ### Author Response · Authors · 2022-08-02
> **Response to Reviewer KcfL [1/2]**
>
> Thank you very much for your time and effort in reviewing our work. We are very glad to know you find our problem setting is interesting, the idea is appealing, the proposed approach is novel and well-structured, and the paper is well-written and covers lots of baselines on several synthetic and real-world problems.
>
> We address your concerns as follows.
>
> > **1. Practicality of the Approximate Pareto Set**
>
> This is an important concern for the usage of the approximate Pareto set. We want to tackle it from two perspectives: 1) how can the decision-maker use the approximate Pareto set, and 2) what if the budget is not enough to get a good approximate Pareto set.
>
> **How can a decision-maker use an approximate Pareto set?**
>
> We fully agree with the reviewer that it could be very risky, especially in those safety-critical applications, to only rely on an approximate Pareto set to make the final decision. We have added more discussion on this issue in Appendix B for limitation and Appendix C for potential societal impact.
>
> On the other hand, with our proposed method, decision-makers can have both the evaluated solutions and the approximate Pareto set at the same time. The approximate Pareto set can provide extra useful information to better support their decision-making. More specifically,
>
> - It can help the decision-makers to better understand the (approximate) trade-offs they will make when they choose an already evaluated solution as their final solution. The approximate Pareto set and the corresponding surrogate objective values around the chosen solution can provide valuable information to understand what we can gain or lose by adjusting the current chosen solution.
> - It allows the decision-maker to explore the whole approximate Pareto set/front easily. If none of the already evaluated solutions can precisely satisfy the decision-maker's preferred trade-off (which could be possible for applications with infinite Pareto sets), the decision-maker can rely on the approximate Pareto set/front to choose the preferred solution for further evaluation.
> - When the optimization modeler and the final decision-maker are not the same person, the approximate Pareto set/front provides a much more efficient way for them to communicate and discuss the whole trade-offs among different objectives during or post the optimization process. This demand is common and important for many applications [1].
> - Finally, an important application for MOBO is to help domain experts efficiently conduct experiments. The approximate Pareto set/front might contain useful patterns and structures which can help the domain expert obtain more useful information from the experiment. It also provides a way for domain experts to incorporate their knowledge into the optimization process (e.g., choose the most concerned region, and eliminate some uninteresting locations). The proposed Pareto set learning method could be a novel approach to support "bring decision maker back into the loop" for Bayesian optimization [2].
>
> We have added the above discussion on the practicality of approximate Pareto set in Appendix A.3 of the revised paper.
>
> **What if the budget is insufficient to obtain a good approximate Pareto set?**
>
> The quality of the approximate Pareto set heavily depends on the quality of the surrogate models. If the evaluation budget is insufficient to build a good set of surrogate models (for each objective), we cannot obtain a good approximate Pareto set. This is also a challenge for the general (multi-objective) Bayesian optimization methods.
>
> One possible method to address this challenge is to leverage the user preference information into the expensive optimization process, such as in [3] as pointed out by the reviewer. In this way, the general MOBO method can spend the limited evaluation budget mainly on the user-preferred region rather than the whole decision space. Similarly, our proposed method can only build a partial approximate Pareto set with the user-preferred trade-offs. This is also related to the following two concerns raised by the reviewer.

---

### Author Response · Authors · 2022-08-02
**Response Summary**

We sincerely thank all reviewers for their constructive comments and valuable suggestions. Following their suggestions, we have carefully revised our paper and the major revision can be summarized as follow:

- **New Appendix A** to discuss motivations of the proposed PSL method, and practicality of the approximate Pareto set.
- **New Appendix B** to discuss the limitations of the proposed method in details, and also provide some potential future research direction.
- **New Appendix C** to discuss potential societal impact.
- **6 New Test Problems** with various shapes of Pareto sets have been proposed in Appendix E.2 and tested in Appendix F.1 with 6 and 20 dimensional search space.
- **More Experiments** have been conducted to further analyze the proposed PSL method. Those results can be found in Appendix F.
- **Many Careful Modifications** have been made for better exposition and clarification.

We combine the main paper and the appendix into a single file but keep the main paper 9-page following this year's requirement. We will adjust the content once an extra page is allowed. All major changes are highlighted in **orange**.

Point-to-point responses can be found below to each reviewer. We are also glad to continually improve our work to address any further concern.

Paper5978 Authors


#################################  **Revision Aug 09**  #################################

We have further revised the main paper and added multiple new experiments in the Appendix. The major change in this revision is highlighted in **blue**:

- **Further Careful Modifications** to the main paper have been made for better exposition and clarification.
- **Four New Experiments** have been conducted in Appendix F.11-F.14 to further analyze the proposed PSL method.


Paper5978 Authors

---

### Meta-Review · Area_Chair_xoNC · 2022-08-27

**Recommendation:** Accept
**Confidence:** Certain

**Metareview:**

This paper studied the problem of (batch) multi-objective Bayesian optimization (BO). It considers a novel perspective to solve problems with an infinite size pareto optimal set by finding a pareto manifold of solutions. The acquisition strategy uses Chebyshev scalarization. The key idea is to learn a mapping from preferences (i.e., scalarization parameters) to the Pareto optimal solution and use it to guide the acquisition strategy and BO process to approximate the Pareto set. Experimental results demonstrate the effectiveness of the proposed approach.

All reviewers agreed about the novel perspective from which the multi-objective BO problem was studied, but also raised some concerns and questions. The authors gave satisfactory responses to most of the review comments and revised the paper to improve it. Two reviewers strongly supported accepting the paper and two of them gave borderline accept. Authors satisfactorily addressed the main concern of one of them (i.e., test problems are too simple). Some of the comments from Reviewer MK65 needs further work, which is acknowledged by the authors.

The overall approach is novel, advances scalarization based multi-objective BO, produced good results, and has the potential to generate good interest in the BO community. Therefore, I recommend acceptance.

**Award:**

No

---

### Decision · Program_Chairs · 2022-09-14

Accept